# Open-World LLM Logical Reasoning

**Ye Mo** [1]  **Chuan Zhou** [2]  **Fengxiang Cheng** [3]  **Jialin Yu** [4]  **Liangming Pan** [5 6]  **Fenrong Liu** [7]
**Sheng Zhou** [1 ✉]  **Haoxuan Li** [2 4 ✉]  **Zhouchen Lin** [2 8 ✉]  **Philip Torr** [4]

## Abstract

Large Language Models (LLMs) achieve remarkable performance but struggle with complex logical reasoning, particularly in real-world settings. Existing research is largely confined to the closed-world assumption, which posits that all premises required for reasoning are explicitly provided. However, real-world tasks frequently exhibit open-world characteristics, where the provided information is insufficient to infer a conclusion due to missing premises or implicit commonsense knowledge. To address this, we propose OpenIKLR, an Open-world Incomplete-Knowledge-aware Logical Reasoning framework that integrates symbolic logic solvers with LLMs. OpenIKLR first translates natural language into symbolic representations to pinpoint reasoning gaps via a logical solver. It then iteratively generates a minimal set of necessary missing premises using LLMs. To ensure these added premises are both logically sound and factually accurate, we introduce a dual-verification: logic verification via the solver and fact verification via the LLMs. Experiments show that OpenIKLR consistently outperforms existing logical reasoning and RAG baselines across multiple backbones and real-world datasets. The code is available at https://github.com/OpenIKLR/OpenIKLR.

[1]Zhejiang Key Laboratory of Accessible Perception and Intelligent Systems, Zhejiang University, Hangzhou, China [2]State Key Lab of General AI, School of Intelligence Science and Technology, Peking University, Beijing, China [3]Institute for Logic, Language and Computation, University of Amsterdam, Amsterdam, Netherlands [4]Department of Engineering Science, University of Oxford, Oxford, United Kingdom [5]State Key Laboratory of Multimedia Information Processing, School of Computer Science, Peking University, Beijing, China [6]Beijing Academy of Artificial Intelligence, Beijing, China [7]The Tsinghua-UvA JRC for Logic, Department of Philosophy, Tsinghua University, Beijing, China [8]Institute for Artificial Intelligence, Peking University, Beijing, China. Correspondence to: Sheng Zhou <zhousheng_zju@zju.edu.cn>, Haoxuan Li <hxli@stu.pku.edu.cn>, Zhouchen Lin <zlin@pku.edu.cn>.

*Proceedings of the 43rd International Conference on Machine Learning*, Seoul, South Korea. PMLR 306, 2026. Copyright 2026 by the author(s).

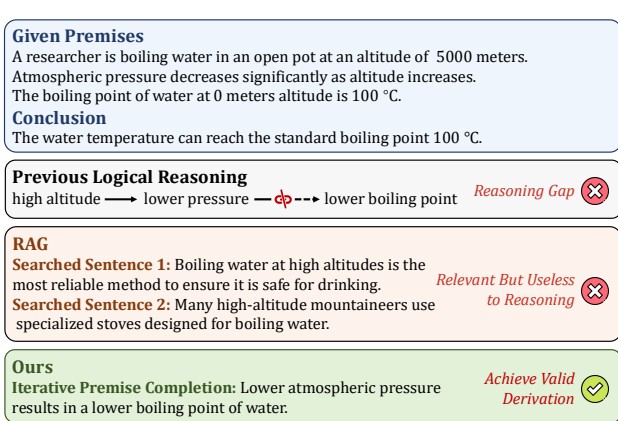

**Given Premises**
A researcher is boiling water in an open pot at an altitude of 5000 meters.
Atmospheric pressure decreases significantly as altitude increases.
The boiling point of water at 0 meters altitude is 100 ℃.
**Conclusion**
The water temperature can reach the standard boiling point 100 ℃.

**Previous Logical Reasoning**
high altitude ⟶ lower pressure ⟶ lower boiling point    *Reasoning Gap* ✗

**RAG**
**Searched Sentence 1:** Boiling water at high altitudes is the most reliable method to ensure it is safe for drinking.    *Relevant But Useless to Reasoning* ✗
**Searched Sentence 2:** Many high-altitude mountaineers use specialized stoves designed for boiling water.

**Ours**
**Iterative Premise Completion:** Lower atmospheric pressure results in a lower boiling point of water.    *Achieve Valid Derivation* ✓

*Figure 1.* A motivating example: a reasoning gap in an open-world scenario where implicit commonsense knowledge is required to derive a logical conclusion.

## 1. Introduction

Large Language Models (LLMs) have demonstrated impressive capabilities across diverse natural language processing tasks, but still struggle with complex logical reasoning, which significantly limits their applicability in real-world scenarios (Yang et al., 2025b; Chen et al., 2026; Meng et al., 2026). Logical reasoning tasks usually require LLMs to determine whether a statement can be logically deduced from the given information using formal logical rules (Xu et al., 2024b; Cheng et al., 2025; Lin et al., 2025). Numerous efforts have been made to enhance the logical reasoning capabilities of LLMs, which generally follow a two-stage framework: translating natural language (NL) problems into symbolic language (SL) and then performing reasoning using these symbolic representations (Feng et al., 2024; 2026; Yang et al., 2026a). In general, the NL-to-SL translation stage primarily leverages LLM prompting, while subsequent logical reasoning can be executed via external logic solvers (Ye et al., 2023; Olausson et al., 2023), further LLM prompting (Xu et al., 2024a; Liu et al., 2025d; Li et al., 2024; Kesseli et al., 2026), or specially fine-tuned models (Wan et al., 2024; Morishita et al., 2024; Feng et al., 2026).

Most existing logical reasoning methods are limited to the **Closed-World Reasoning**, which presumes that all premises required for deriving the conclusion are already explicitly given in the context (Ryu et al., 2025; Tafjord

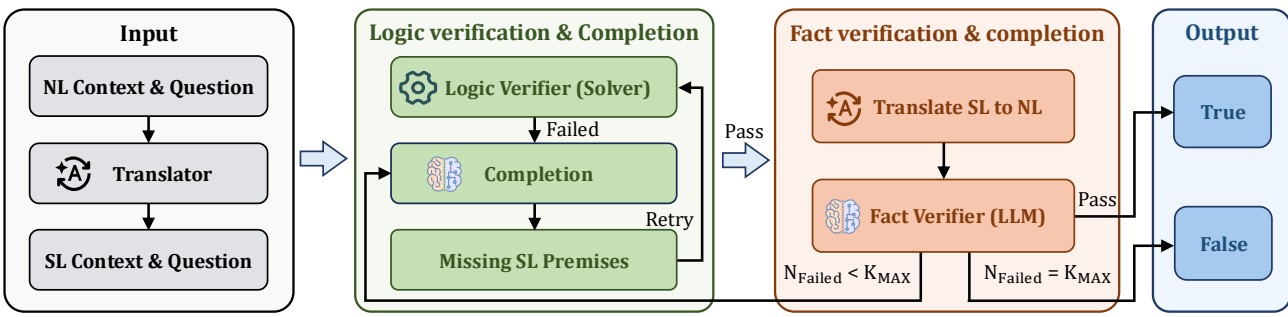

*Figure 2.* The OpenIKLR framework architecture, depicting the iterative flow between symbolic translation, logic completion, and fact verification. (1) Input & Translation: Maps natural language context and questions into symbolic representations to enable precise formal modeling. (2) Logic Verification and Completion: Employs a symbolic solver to pinpoint reasoning gaps and iteratively generates the minimal set of missing premises required for the proof. (3) Fact Verification and Completion: Validates the factual accuracy of generated premises via an LLM to ensure the final output is grounded in real-world knowledge. Pseudocode of the algorithm is in Appendix B.

et al., 2021; Tian et al., 2026). However, real-world tasks require **Open-World Reasoning**, in which the provided information is insufficient to infer a conclusion due to missing premises or implicit commonsense knowledge. This requires LLMs to actively explore the missing information needed for logically deducing the conclusion.

A motivating example is shown in Figure 1, in which the logical reasoning chain should be "high altitude → lower pressure → lower boiling point". However, previous logical reasoning methods can only obtain the reasoning chain "high altitude → lower pressure", but fail to reach the final conclusion given the three premises since missing the information that "lower pressure results in lower boiling point". Moreover, we find that even the state-of-the-art Retrieval-Augmented Generation (RAG) method still fails to reason correctly. This is because (i) RAG relies on statistical correlation and keyword matching, resulting in retrieved information relevant but usually useless to reasoning, and (ii) RAG is constrained by the given external knowledge base, failing to achieve open-world reasoning.

To fill the reasoning gap in open-world scenarios, we propose OpenIKLR, an **Open**-world **I**ncomplete-**K**nowledge-aware **L**ogical **R**easoning framework, which integrates the formal logical reasoning via symbolic solvers and the semantic reasoning via world knowledge of LLMs. As shown in Figure 2, OpenIKLR first translates the context of the logical reasoning problem into symbolic forms, allowing the logical solver to check whether there are any missing logical premises in the reasoning sample. If the solver fails to deduce the target conclusion, we then employ an iterative completion mechanism to generate the minimal set of missing premises. To ensure both logical rigor and factual correctness of added premises, we introduce a dual-verification process: the symbolic solver verifies that combining existing and newly added premises logically deduces the conclusion, while the LLM validates the correctness of the newly generated premises using its world knowledge.

Our contributions are summarized as follows:

- We propose an open-world reasoning task in which the provided information is insufficient to infer a conclusion, requiring LLMs to actively explore the missing knowledge needed for logical deduction.

- We introduce a solver-guided completion framework that identifies minimal missing information based on logical necessity rather than keyword relevance, with a dual-verification process ensuring the added knowledge is both logically sound and factually accurate.

- We conduct extensive experiments showing that our OpenIKLR consistently outperforms state-of-the-art logical reasoning and RAG-based baselines across multiple backbones and datasets.

## 2. Problem Setup

We formalize the task of open-world logical reasoning under incomplete knowledge. Let $P$ be a set of natural language premises and $C$ be a natural language conclusion. In the symbolic space, let $\Gamma$ represent the set of translated symbolic premises and $\phi$ represent the symbolic conclusion.

The goal of closed-world reasoning is to determine if $\Gamma \vDash \phi$. Extensive related work exists, with detailed discussions provided in Appendix A. However, in our proposed open-world scenarios, the provided information can be insufficient for deduction:

$$\Gamma \nvDash \phi,$$

where the symbol $\nvDash$ represents logical insufficiency, meaning that the conclusion $\phi$ cannot be deduced from the initial set $\Gamma$ due to a "reasoning gap" caused by missing premises. Our objective is to find a **minimal set of missing premises** $\Delta$, such that the proof becomes valid:

$$\Gamma \cup \Delta \vDash \phi,$$

where the symbol $\vDash$ denotes logical entailment, indicating that the conclusion $\phi$ can be strictly deduced from the set of premises $\Gamma \cup \Delta$ using formal logical rules. For $\Delta$ to be a valid completion, it must satisfy two primary constraints: (1) Logical Sufficiency: The union of the original premises and the missing knowledge must allow the solver to successfully reach the conclusion $\phi$. (2) Factual Accuracy: Every premise $p' \in \Delta$ generated by the LLM must be a factually true statement consistent with real-world knowledge.

## 3. Methodology

The OpenIKLR framework is designed to address the "reasoning gap" in open-world scenarios where provided natural language premises $\mathcal{P}$ are insufficient to derive a target conclusion $\mathcal{C}$. Unlike traditional closed-world settings that assume all necessary information is explicitly provided, OpenIKLR operates under the premise that real-world reasoning often relies on implicit commonsense or missing context. The framework integrates the formal rigor of symbolic logic solvers with the generative world knowledge of LLMs to identify and complete these missing links.

As the pipeline illustrated in Figure 2 shows, the premises in natural language (NL) are translated to symbolic language (SL) to facilitate precise gap identification. The logic verification and completion process is inherently iterative: the framework first leverages a logic solver to verify whether the premises are sufficient to derive the conclusion. If the proof fails, it invokes an LLM to generate a minimal set of missing premises $\Delta = (\mathcal{P}', \mathcal{P}'_s)$, which are then factually checked by the LLM. Based on whether the completed premises pass the logic verifier and fact verifier, we can infer the final answer. This dual-verification loop ensures that the final reasoning chain is both logically sound and factually grounded.

### 3.1. Symbolic Parsing

The first stage of the pipeline involves a TRANSLATOR module that maps the set of NL premises $\mathcal{P}$ and the conclusion $\mathcal{C}$ into their corresponding symbolic representations $(\mathcal{P}_s, \mathcal{C}_s)$. This formalization is essential to move beyond the statistical keyword matching of standard retrieval approaches and into the domain of strict logical entailment.

As shown in Figure 3(a), a categorical statement such as "All squares are four-sided things" is translated into first-order logic as $\forall x(\text{Square}(x) \rightarrow \text{FourSided}(x))$. By representing the problem symbolically, the framework prepares the input for rigorous verification where the objective is to satisfy the entailment $\mathcal{P}_s \vDash \mathcal{C}_s$. Precise parsing is critical, as any error in the initial SL representation would lead to incorrect gap identification in subsequent steps.

### 3.2. Missing Premises Completion

In open-world settings, the symbolic solver typically finds that the conclusion may not be derived from the initial set, denoted as $\mathcal{P}_s \nvDash \mathcal{C}_s$. When the solver returns a failure status $\mathcal{V}$ or a result $\mathcal{R}$ that is not True, the COMPLETION module is triggered. This module generates the minimal set of missing premises based on the symbolic language $(\mathcal{P}_s, \mathcal{C}_s)$, and the failed proof trace $\mathcal{T}$.

As exemplified in Figure 3(b), given a context where "The nail is made of iron", the solver initially fails to conclude "Can a nail conduct electricity?" because intermediate relationships—such as the fact that iron is a metal and metals conduct electricity—are missing. The completion module generates these symbolic missing links, $\mathcal{P}'_s = \{\text{IsMetal}(\text{iron}), \text{ConductsElectricity}(\text{metal})\}$, along with their natural language counterparts $\mathcal{P}'$. This targeted generation ensures that the added knowledge is logically sufficient to bridge the gap and satisfy the proof $\mathcal{P}_s \cup \mathcal{P}'_s \vDash \mathcal{C}_s$.

### 3.3. Solver Verification

Once new symbolic premises $\mathcal{P}'_s$ are proposed, the augmented premises set in symbolic representation is resubmitted to the SOLVER to verify the logical validity of the expanded reasoning chain. The solver outputs a new execution status $\mathcal{V}$, a boolean result $\mathcal{R}$, and a formal proof $\mathcal{T}$ based on the union $\mathcal{P}_s \cup \mathcal{P}'_s$.

The framework is governed by an iterative loop that persists while the iteration count $k < K_{\max}$. As seen in the transition within Figure 3(b), the system may require multiple retries to refine symbolic premises until a valid proof is constructed. This iterative symbolic verification prevents logical "hallucinations" by forcing the LLM to provide premises that strictly conform to the deductive rules of the logic solver. If the solver confirms the logic ($\mathcal{V}$ is successful and $\mathcal{R}$ is True), the framework increments the true count $n$ and proceeds to the final factual grounding stage.

### 3.4. Fact Verification

Logical validity alone does not guarantee truth; a reasoning chain can be valid while relying on factually false premises. To prevent this, OpenIKLR first translates $\mathcal{P}'_s$ to natural language $\mathcal{P}'$, and then employs a FACTVERIFIER module where the LLM verifies the natural language versions of the generated premises $\mathcal{P}'$ based on real-world knowledge.

As shown in Step 5.1 of Figure 3(b), the system verifies the factual truth $\mathcal{F}$ of generated statements such as "Metal can conduct electricity" and "Iron is Metal". If $\mathcal{F}$ is True, the framework immediately returns True for the predicted label $\mathcal{Y}$. If any premise in $\mathcal{P}'$ is flagged as factually incorrect, that iteration is discarded, and the system continues

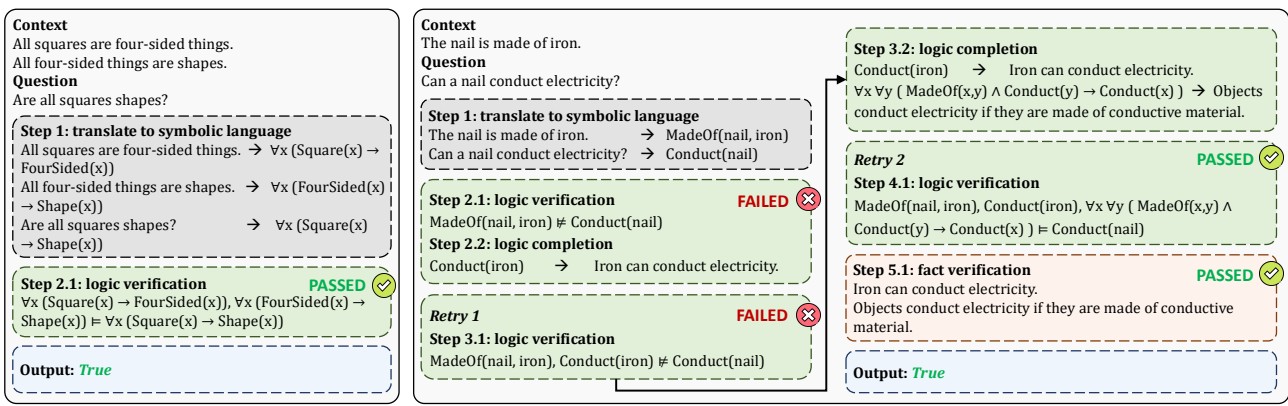

**(a)** Example of symbolic translation ($\mathcal{P}_s$, $\mathcal{C}_s$) and logic verification for a closed-world reasoning task.

**(b)** Initial stage of open-world reasoning: the solver identifies a logical gap and triggers premise completion. Final verification stage: missing premises are validated for both logical sufficiency and factual accuracy.

*Figure 3.* Procedural examples of iterative completion for the missing premises.

the loop to seek another logical path. This dual-constraint mechanism—logical sufficiency via the symbolic solver and factual accuracy via the LLM—ensures the final output is both logically rigorous and grounded in reality. In cases where the loop completes without a successful fact-verified proof, the system evaluates the cumulative results to determine the final label, often defaulting to a Chain-of-Thought ($\mathcal{Y}_{CoT}$) fallback if the threshold $n > K_{\max}/2$ is not met.

### 3.5. Theoretical Analysis

Let $\mathcal{D}$ be a distribution over reasoning instances $(\Gamma, \phi, y)$ and let $S = \{(\Gamma_i, \phi_i, y_i)\}_{i=1}^n \sim \mathcal{D}^n$. Let $\mathcal{P}$ be a finite premise space. We consider randomized predictors defined by a posterior distribution $Q$ over premise sets $\Delta$. For any premise set $\Delta \subseteq \mathcal{P}$, define the deterministic predictor $f_\Delta(\Gamma, \phi) = \mathbf{1}[\Gamma \cup \Delta \vDash \phi]$. We consider LLM as a randomized predictor induced by a distribution $Q$ over premise sets $\Delta$. The LLM predictor draws $\Delta \sim Q$ and our method predicts using $f_\Delta$. We define the true risk and empirical risk of $f_\Delta$ as $R_\mathcal{D}(f_\Delta)$ and $\hat{R}_S(f_\Delta)$. For a posterior $Q$ over $\Delta$, define the Gibbs risks:

$$R_\mathcal{D}(Q) = \mathbb{E}_{\Delta \sim Q} R_\mathcal{D}(f_\Delta), \qquad \hat{R}_S(Q) = \mathbb{E}_{\Delta \sim Q} \hat{R}_S(f_\Delta).$$

Let $P$ be a data-independent prior distribution over premise sets $\Delta \subseteq \mathcal{P}$. For example, the prior distribution reflects our preference for premise sets with smaller cardinality. The posterior $Q$ may depend arbitrarily on the sample $S$, e.g., through the specific premise completion process. We choose a prior that penalizes larger premise sets:

$$P(\Delta) \propto \exp(-\alpha|\Delta|), \alpha > 0.$$

**Theorem 3.1** (PAC-Bayes Bound)**.** *For any prior $P(\Delta)$ and posterior $Q(\Delta)$ over the missing premises $\Delta$ and any $\delta \in (0,1)$, with probability at least $1 - \delta$,*

$$R_\mathcal{D}(Q(\Delta)) \leq \hat{R}_S(Q(\Delta)) + \sqrt{\frac{\mathrm{KL}(Q(\Delta)\|P(\Delta)) + \log\frac{1}{\delta}}{2(n-1)}}.$$

**Theorem 3.2** (Monotonicity)**.** *Fix $\alpha > 0$, $n$, and $\delta$. Consider $Q_s$ and $Q_l$ such that $\mathbb{E}_{Q_s}[|\Delta|] \leq \mathbb{E}_{Q_l}[|\Delta|]$,*

$$\mathbb{E}_{Q_l}[|\Delta|] - \mathbb{E}_{Q_s}[|\Delta|] \geq \frac{\mathbb{E}_{Q_s}[\log Q_s(\Delta)] - \mathbb{E}_{Q_l}[\log Q_l(\Delta)]}{\alpha},$$

*then the PAC-Bayes upper bound for $Q_s$ is smaller than $Q_l$.*

The condition in Theorem 3.2 is easily satisfied as the prior penalizes $\mathbb{E}_{Q_s}$ tends to shorter premise sets, often leading to higher confidence, which in turn causes $\mathbb{E}_{Q_s}[\log Q_s(\Delta)]$ to be extreme and less than $\mathbb{E}_{Q_l}[\log Q_l(\Delta)]$. We show that the bound increases as the number of premises completed grows. This provides a principled justification for solver-guided minimal premise completion. A detailed proof is provided in Appendix C.

Additionally, we provide a time complexity analysis of our algorithm. The complexity consists of the inference cost of LLM and the computational cost of the logic solver. Let $T_{LLM}(l)$ be the LLM inference time cost for context with length $l$, which is theoretically $O(l^2)$ (Keles et al., 2023). Let $T_{SOL}(n)$ be the worst-case time complexity of the logic solver under $n$ premises. For instance, the complexity of a first-order logic solver is typically $O(2^n)$. Denote the length of the translation prompt as $l_{init}$, and $l_k$ for the prompt of the $k$-th missing premise completion iteration. Overall, the time complexity is $O(l_{init}^2 + \sum_{k=1}^{K_{\max}}[2l_k^2 + 2^{|\mathcal{P}_s \cup \mathcal{P}_s'|}])$.

## 4. Experiments

In this section, we investigate three research questions and provide in-depth analyses (Sec. 4.5) to extensively evaluate the performance of OpenIKLR.

**RQ 1:** How does OpenIKLR perform across different models, datasets, and model scales? (Sec. 4.2)

**RQ 2:** How does each component of OpenIKLR contribute to its overall performance? (Sec. 4.3)

*Table 1.* Comparison of different LLMs and methods on the **Multi-LogiEval** benchmark. The percentages indicate the proportion of open-world samples in each subset. **Bold** indicates the best performance within each model.

| MODEL | METHOD | 90% | | | 70% | | | 50% | | |
|---|---|---|---|---|---|---|---|---|---|---|
| | | ACC. | RECALL | F1 | ACC. | RECALL | F1 | ACC. | RECALL | F1 |
| DEEPSEEK-V3.2 | DIRECT | 42.06 | 33.57 | 48.80 | 48.07 | 38.73 | 53.99 | 53.57 | 46.84 | 62.17 |
| | CoT | 50.64 | 42.78 | 58.78 | 58.51 | 50.18 | 65.56 | 60.43 | 54.04 | 68.98 |
| | LINC | 20.74 | 24.17 | 33.41 | 23.03 | 26.91 | 35.49 | 28.00 | 32.46 | 42.33 |
| | LOGICLM | 33.62 | 32.70 | 44.76 | 38.34 | 37.82 | 49.11 | 43.86 | 46.49 | 57.42 |
| | CR | 41.49 | 32.00 | 47.36 | 51.07 | 41.45 | 57.14 | 53.29 | 48.42 | 62.80 |
| | DETERMLR | 33.14 | 21.15 | 34.28 | 41.27 | 28.88 | 43.71 | 47.11 | 39.29 | 54.81 |
| | SYMBCOT | 34.19 | 24.87 | 38.34 | 41.63 | 31.27 | 45.74 | 41.86 | 36.67 | 50.67 |
| | HYDE | 37.91 | 28.35 | 42.89 | 45.64 | 36.12 | 51.16 | 45.71 | 37.83 | 53.20 |
| | HYDE+OURS | 59.71 | 60.00 | 70.99 | 63.57 | 64.61 | 73.63 | 66.10 | 67.95 | 76.53 |
| | SIRERAG | 39.00 | 28.35 | 43.29 | 46.57 | 35.57 | 51.17 | 51.57 | 43.43 | 59.33 |
| | SIRERAG+OURS | 61.09 | 60.87 | 72.02 | 64.86 | 65.70 | 74.64 | 68.05 | 70.40 | 78.21 |
| | **OURS** | **68.53** | **69.39** | **78.39** | **71.67** | **73.09** | **80.24** | **72.71** | **75.09** | **81.76** |
| GPT-5 | DIRECT | 52.36 | 43.83 | 60.22 | 59.08 | 49.82 | 65.71 | 63.86 | 57.37 | 72.11 |
| | CoT | 52.79 | 45.04 | 61.08 | 61.09 | 52.91 | 68.15 | 65.14 | 59.12 | 73.42 |
| | LINC | 15.02 | 18.09 | 25.94 | 17.45 | 21.09 | 28.68 | 24.71 | 28.25 | 37.93 |
| | LOGICLM | 29.18 | 29.91 | 41.00 | 33.48 | 35.82 | 45.87 | 42.71 | 46.32 | 56.84 |
| | CR | 35.19 | 22.43 | 36.29 | 41.77 | 27.09 | 42.27 | 44.86 | 34.04 | 50.13 |
| | DETERMLR | 47.07 | 38.43 | 54.43 | 53.65 | 44.91 | 60.39 | 58.14 | 52.28 | 67.04 |
| | SYMBCOT | 41.34 | 31.48 | 46.89 | 50.07 | 40.00 | 55.77 | 55.14 | 49.30 | 64.16 |
| | HYDE | 33.49 | 25.76 | 39.14 | 35.13 | 27.11 | 40.15 | 38.06 | 31.50 | 46.61 |
| | HYDE+OURS | 53.79 | 56.87 | 66.94 | 55.22 | 61.52 | 68.42 | 58.57 | 63.22 | 71.34 |
| | SIRERAG | 40.77 | 32.00 | 47.06 | 46.49 | 37.02 | 52.17 | 50.57 | 43.61 | 59.00 |
| | SIRERAG+OURS | 58.66 | 62.09 | 71.19 | 57.37 | 61.34 | 69.40 | 61.86 | 66.20 | 73.90 |
| | **OURS** | **66.95** | **72.35** | **78.27** | **68.24** | **75.09** | **78.82** | **72.43** | **79.12** | **82.37** |
| GEMINI-2.5-FLASH | DIRECT | 49.36 | 42.26 | 57.86 | 57.22 | 51.09 | 65.27 | 63.00 | 58.07 | 71.88 |
| | CoT | 50.64 | 45.22 | 60.12 | 56.65 | 52.00 | 65.37 | 63.43 | 60.70 | 73.00 |
| | LINC | 16.02 | 19.13 | 27.26 | 17.02 | 20.36 | 27.86 | 23.71 | 27.37 | 36.88 |
| | LOGICLM | 29.04 | 29.74 | 40.81 | 32.47 | 34.18 | 44.34 | 42.00 | 45.61 | 56.16 |
| | CR | 49.07 | 43.30 | 58.31 | 56.51 | 50.36 | 64.57 | 58.86 | 54.56 | 68.35 |
| | DETERMLR | 38.38 | 27.11 | 42.45 | 39.79 | 28.57 | 44.44 | 49.22 | 41.10 | 57.76 |
| | SYMBCOT | 42.92 | 33.74 | 49.30 | 53.36 | 44.00 | 59.75 | 57.86 | 51.58 | 66.59 |
| | HYDE | 37.05 | 26.61 | 41.02 | 43.92 | 33.94 | 48.83 | 49.00 | 42.21 | 57.45 |
| | HYDE+OURS | 61.98 | 63.04 | 74.07 | 67.28 | 69.89 | 78.43 | 69.71 | 72.14 | 80.59 |
| | SIRERAG | 49.00 | 39.48 | 55.98 | 55.71 | 45.55 | 61.82 | 61.40 | 54.64 | 69.72 |
| | SIRERAG+OURS | 63.57 | 69.41 | 65.80 | 66.38 | 69.75 | 74.37 | 70.09 | 77.52 | 80.40 |
| | **OURS** | **67.24** | **73.04** | **78.58** | **67.95** | **75.64** | **78.79** | **72.00** | **79.82** | **82.28** |
| CLAUDE-3.5-HAIKU | DIRECT | 37.34 | 28.17 | 42.52 | 42.20 | 32.73 | 47.12 | 43.00 | 35.96 | 50.68 |
| | CoT | 40.34 | 32.87 | 47.55 | 46.21 | 39.45 | 53.58 | 44.43 | 38.60 | 53.08 |
| | LINC | 17.02 | 20.17 | 28.57 | 18.31 | 21.82 | 29.59 | 24.14 | 27.72 | 37.31 |
| | LOGICLM | 29.76 | 29.22 | 40.63 | 34.05 | 34.73 | 45.31 | 38.29 | 39.65 | 51.13 |
| | CR | 46.72 | 36.52 | 52.90 | 51.78 | 41.82 | 57.57 | 51.56 | 46.14 | 60.67 |
| | DETERMLR | 39.50 | 18.34 | 30.46 | 43.33 | 18.92 | 31.00 | 40.38 | 20.21 | 33.05 |
| | SYMBCOT | 46.07 | 38.09 | 53.74 | 52.93 | 45.09 | 60.12 | 53.86 | 48.25 | 63.00 |
| | HYDE | 29.47 | 16.35 | 27.61 | 34.48 | 19.60 | 32.05 | 35.00 | 22.24 | 35.83 |
| | HYDE+OURS | 34.19 | 24.17 | 37.67 | 39.48 | 29.76 | 43.68 | 39.71 | 33.10 | 47.25 |
| | SIRERAG | 36.00 | 29.04 | 42.71 | 40.86 | 31.22 | 45.38 | 40.03 | 31.35 | 45.96 |
| | SIRERAG+OURS | 43.71 | 41.39 | 54.71 | 48.50 | 46.28 | 58.55 | 48.93 | 47.99 | 60.42 |
| | **OURS** | **62.37** | **66.09** | **74.29** | **63.23** | **70.36** | **75.07** | **63.43** | **69.12** | **75.48** |

**RQ 3:** How sensitive is OpenIKLR to different hyperparameter settings? (Sec. 4.4)

## 4.1. Settings

**Model.** We evaluate our method across a diverse set of LLMs covering multiple families and scales. Specifically, we include DeepSeek-V3.2 (Liu et al., 2025a), GPT-5 (OpenAI, 2026), Gemini-2.5-Flash (Comanici et al., 2025), Claude-3.5-Haiku (Anthropic, 2026), and Qwen3-235B-A22B (Yang et al., 2025a) as representative SOTA models. To further investigate the effect of model scale, we additionally experiment with smaller variants, Qwen3-32B and Qwen3-8B (Yang et al., 2025a).

**Baseline.** We compare our method against a broad range of baselines spanning different reasoning paradigms, including Direct QA, CoT (Wei et al., 2022), and logical reasoning methods such as CR (Zhang et al., 2024a), LINC (Olausson et al., 2023), DetermLR (Sun et al., 2024), SymbCoT (Xu et al., 2024b), and LogicLM (Pan et al., 2023). Following prior work, we adopt FOL as the symbolic representation language and Prover9 (McCune, 2009) as the theorem prover. In addition, we incorporate RAG-based approaches, including SiReRAG (Zhang et al., 2026) and HyDE (Gao et al., 2023), to assess the effectiveness of external knowledge retrieval in open-world logical reasoning scenarios.

**Dataset.** We use two multi-step logical reasoning benchmarks, FOLIO (Han et al., 2024) and Multi-LogiEval (Patel et al., 2024), as the original data sources. To construct open-world evaluation settings with incomplete knowledge, we employ GPT-4.1 to determine whether each premise in the original datasets is real-world knowledge or common sense. Premises identified as such are subsequently masked, resulting in incomplete samples, which constitute our final test sets. To simulate real-world open-world scenarios, we con-

*Table 2.* Comparison of different LLMs and methods on the **FOLIO** benchmark. The percentages indicate the proportion of open-world samples in each subset. **Bold** indicate best performance within each model.

| MODEL | METHOD | 90% | | | 70% | | | 50% | | |
|---|---|---|---|---|---|---|---|---|---|---|
| | | ACC. | RECALL | F1 | ACC. | RECALL | F1 | ACC. | RECALL | F1 |
| DEEPSEEK-V3.2 | DIRECT | 85.14 | 79.94 | 85.49 | 85.23 | 82.17 | 85.30 | 85.77 | 82.61 | 86.04 |
| | CoT | 83.36 | 76.99 | 83.52 | 85.68 | 81.01 | 85.91 | 86.54 | 82.61 | 86.69 |
| | LINC | 23.75 | 29.50 | 29.76 | 31.82 | 38.40 | 37.76 | 40.00 | 45.65 | 44.68 |
| | LOGICLM | 45.56 | 50.44 | 50.37 | 51.14 | 57.81 | 56.03 | 58.08 | 63.77 | 61.75 |
| | CR | 71.24 | 61.95 | 70.23 | 75.23 | 68.78 | 74.94 | 81.15 | 77.54 | 81.37 |
| | DETERMLR | 78.10 | 61.19 | 75.37 | 82.11 | 68.22 | 80.50 | 85.66 | 74.45 | 84.65 |
| | SYMBCoT | 63.97 | 50.15 | 60.39 | 69.77 | 57.38 | 67.16 | 73.85 | 60.87 | 71.19 |
| | HYDE | 78.68 | 70.50 | 78.36 | 79.09 | 71.73 | 78.70 | 83.08 | 77.54 | 82.95 |
| | HYDE+OURS | 85.81 | 82.01 | 86.34 | 85.97 | 83.54 | 86.46 | 86.64 | 86.96 | 87.27 |
| | SIRERAG | 82.26 | 71.68 | 81.54 | 84.39 | 74.68 | 83.69 | 84.73 | 77.54 | 84.25 |
| | SIRERAG+OURS | 86.59 | 85.84 | 87.52 | 87.66 | 89.45 | 88.45 | 86.97 | 88.41 | 87.77 |
| | **OURS** | **90.81** | **94.12** | **91.82** | **90.27** | **94.54** | **91.28** | **90.46** | **94.96** | **91.35** |
| GPT-5 | DIRECT | 80.01 | 66.96 | 79.21 | 82.57 | 72.57 | 82.49 | 85.12 | 77.54 | 85.43 |
| | CoT | 80.26 | 67.55 | 79.79 | 83.41 | 72.57 | 82.64 | 86.14 | 77.68 | 86.04 |
| | LINC | 20.84 | 26.25 | 26.65 | 27.27 | 31.65 | 31.91 | 34.62 | 39.13 | 38.85 |
| | LOGICLM | 48.14 | 51.62 | 52.16 | 53.18 | 56.96 | 56.72 | 60.00 | 64.49 | 63.12 |
| | CR | 61.97 | 40.39 | 54.41 | 66.59 | 48.37 | 61.34 | 71.80 | 56.25 | 68.35 |
| | DETERMLR | 79.24 | 59.35 | 77.20 | 80.55 | 66.54 | 79.59 | 83.92 | 76.68 | 82.02 |
| | SYMBCoT | 68.34 | 43.17 | 60.30 | 73.32 | 52.19 | 68.19 | 76.11 | 57.78 | 72.56 |
| | HYDE | 77.56 | 71.42 | 77.76 | 79.09 | 73.54 | 79.19 | 79.23 | 73.33 | 79.15 |
| | HYDE+OURS | 80.71 | 75.06 | 79.59 | 81.48 | 80.60 | 81.21 | 81.77 | 82.30 | 82.30 |
| | SIRERAG | 77.98 | 71.71 | 77.26 | 80.32 | 74.87 | 80.43 | 80.41 | 77.32 | 80.62 |
| | SIRERAG+OURS | 81.47 | 78.02 | 77.01 | 81.45 | 78.49 | 79.51 | 83.31 | 82.30 | 83.36 |
| | **OURS** | **84.49** | **87.02** | **86.01** | **86.82** | **87.49** | **88.51** | **88.46** | **91.30** | **89.36** |
| GEMINI-2.5-FLASH | DIRECT | 68.41 | 70.50 | 69.60 | 73.09 | 73.42 | 74.00 | 75.77 | 74.88 | 75.88 |
| | CoT | 69.31 | 70.85 | 70.77 | 73.64 | 73.84 | 75.11 | 76.46 | 75.64 | 76.91 |
| | LINC | 23.75 | 29.20 | 29.55 | 28.86 | 33.33 | 33.55 | 34.62 | 40.58 | 39.72 |
| | LOGICLM | 47.01 | 54.87 | 53.14 | 52.50 | 59.92 | 57.61 | 58.08 | 68.84 | 63.55 |
| | CR | 63.26 | 47.29 | 60.82 | 69.28 | 54.96 | 67.54 | 72.91 | 62.75 | 73.56 |
| | DETERMLR | 68.85 | 55.05 | 71.01 | 72.50 | 58.92 | 73.90 | 74.64 | 61.36 | 75.52 |
| | SYMBCoT | 68.34 | 48.67 | 62.74 | 73.86 | 59.49 | 71.03 | 74.62 | 66.67 | 76.64 |
| | HYDE | 68.05 | 56.34 | 68.83 | 73.41 | 61.18 | 71.25 | 76.15 | 68.12 | 75.20 |
| | HYDE+OURS | 76.32 | 76.36 | 79.41 | 74.04 | 77.94 | 77.94 | 77.32 | 84.74 | 80.91 |
| | SIRERAG | 60.26 | 54.00 | 69.94 | 72.77 | 65.82 | 74.59 | 74.02 | 70.54 | 73.29 |
| | SIRERAG+OURS | 75.46 | 71.44 | 78.88 | 73.29 | 78.07 | 78.02 | 78.95 | 86.48 | 79.75 |
| | **OURS** | **76.58** | **86.73** | **80.22** | **78.41** | **89.45** | **81.70** | **80.54** | **90.58** | **82.39** |
| CLAUDE-3.5-HAIKU | DIRECT | 75.59 | 72.55 | 78.00 | 76.45 | 75.95 | 79.72 | 77.15 | 78.99 | 80.65 |
| | CoT | 75.97 | 75.50 | 77.38 | 76.95 | 82.70 | 79.16 | 77.46 | 81.16 | 80.90 |
| | LINC | 21.16 | 25.66 | 26.28 | 27.50 | 31.65 | 31.98 | 34.62 | 39.86 | 39.29 |
| | LOGICLM | 44.43 | 52.21 | 50.72 | 50.00 | 58.65 | 55.82 | 56.54 | 63.77 | 60.90 |
| | CR | 70.33 | 57.50 | 72.25 | 72.59 | 66.24 | 75.49 | 74.00 | 72.46 | 76.68 |
| | DETERMLR | 73.20 | 61.68 | 72.37 | 76.14 | 67.64 | 75.78 | 76.15 | 68.26 | 75.71 |
| | SYMBCoT | 69.48 | 61.39 | 69.21 | 72.05 | 78.06 | 72.41 | 75.23 | 79.71 | 76.29 |
| | HYDE | 70.60 | 52.21 | 66.04 | 71.82 | 54.43 | 67.54 | 75.00 | 61.59 | 72.34 |
| | HYDE+OURS | 74.15 | 60.47 | 71.93 | 76.59 | 65.82 | 75.18 | 77.38 | 75.36 | 79.31 |
| | SIRERAG | 69.55 | 68.44 | 71.72 | 74.64 | 68.78 | 73.37 | 75.97 | 76.81 | 79.46 |
| | SIRERAG+OURS | 72.58 | 69.65 | 73.33 | 73.30 | 73.12 | 74.19 | 77.45 | 80.58 | 78.34 |
| | **OURS** | **80.29** | **89.38** | **83.24** | **79.09** | **93.25** | **82.77** | **79.62** | **94.20** | **83.07** |

struct multiple dataset variants with different proportions of incomplete instances, specifically 90%, 70%, and 50%. A ratio of 90% indicates that 90% of the samples in the dataset contain incomplete premises. For evaluation, we report Accuracy, Recall, and F1 score as our metrics. Implementation details and prompts are presented in Appendix D and F.

### 4.2. RQ 1: Main Results

This section presents a comprehensive comparison between our method and multiple baselines, including Direct QA, CoT, logical reasoning methods, and RAG-based approaches. As shown in Table 1 and Table 2, OpenIKLR consistently achieves state-of-the-art performance across all evaluated LLM backbones and benchmarks in incomplete knowledge settings. Specifically, on the most challenging 90% open-world setting of both benchmarks, OpenIKLR outperforms the best-performing baselines by a significant

margin. For instance, using the DeepSeek-V3.2 backbone under Multi-LogiEval-90%, our method achieves an Accuracy of 68.53%, whereas logical reasoning methods like LINC and CR only reach 20.74% and 41.49%, respectively.

**Robustness Across Proportions and Model Scales.** OpenIKLR demonstrates remarkable robustness across different "open-world" proportions and model scales. As shown in Table 3, our framework achieves the best performance not only on the massive models like Qwen3-235B-A22B, but also on smaller models (32B and 8B). On Multi-LogiEval (90%), Qwen3-8B gains an accuracy jump from 47.78% (Direct) to 66.85% (Ours). This indicates that OpenIKLR enables smaller models to handle complex reasoning that was previously reserved for larger models. Other results of Qwen3-235B-A22B are in Appendix E.2.

**Superiority over Logic and RAG Baselines.** The experimental results highlight OpenIKLR achieves significant

*Table 3.* Performance of smaller Qwen3 models (235B-A22B, 32B and 8B) on **Multi-LogiEval** and **FOLIO**. The percentages indicate the proportion of open-world samples in each subset. **Bold** indicates the best performance within each model.

| MODEL | METHOD | 90% | | | 70% | | | 50% | | |
|-------|--------|-----|-----|-----|-----|-----|-----|-----|-----|-----|
| | | ACC. | RECALL | F1 | ACC. | RECALL | F1 | ACC. | RECALL | F1 |
| MULTI-LOGIEVAL | | | | | | | | | | |
| QWEN3-235B-A22B | DIRECT | 50.50 | 42.09 | 58.31 | 58.23 | 49.64 | 65.16 | 61.00 | 53.51 | 69.08 |
| | CoT | 52.07 | 45.04 | 60.73 | 57.37 | 50.00 | 64.86 | 63.00 | 57.89 | 71.82 |
| | **OURS** | **69.53** | **72.52** | **79.66** | **69.67** | **74.18** | **79.38** | **73.57** | **79.12** | **82.98** |
| QWEN3-32B | DIRECT | 44.91 | 34.73 | 51.03 | 54.91 | 44.73 | 61.03 | 61.00 | 52.98 | 68.87 |
| | CoT | 46.30 | 36.55 | 52.94 | 56.30 | 56.55 | 62.94 | 60.86 | 56.32 | 70.09 |
| | **OURS** | **68.38** | **72.70** | **79.09** | **68.67** | **75.09** | **79.04** | **69.00** | **75.96** | **79.96** |
| QWEN3-8B | DIRECT | 47.78 | 38.78 | 54.99 | 56.37 | 46.91 | 62.85 | 62.14 | 54.74 | 70.19 |
| | CoT | 51.65 | 44.70 | 60.33 | 58.37 | 51.82 | 66.20 | 63.86 | 59.30 | 72.77 |
| | **OURS** | **66.85** | **67.76** | **77.15** | **72.17** | **72.65** | **80.57** | **77.47** | **79.66** | **85.08** |
| FOLIO | | | | | | | | | | |
| QWEN3-235B-A22B | DIRECT | 80.61 | 66.08 | 78.87 | 78.55 | 74.26 | 78.81 | 84.54 | 76.81 | 81.83 |
| | CoT | 76.09 | 69.03 | 75.97 | 78.41 | 74.26 | 78.75 | 82.69 | 78.99 | 82.89 |
| | **OURS** | **85.10** | **92.44** | **87.24** | **84.62** | **93.31** | **86.77** | **88.89** | **97.84** | **90.37** |
| QWEN3-32B | DIRECT | 83.36 | 71.98 | 82.57 | 84.27 | 78.75 | 84.10 | 82.23 | 82.61 | 87.06 |
| | CoT | 83.42 | 71.39 | 82.80 | 84.32 | 78.06 | 84.28 | 87.31 | 84.78 | 87.64 |
| | **OURS** | **89.68** | **89.68** | **90.42** | **88.67** | **88.97** | **89.19** | **90.97** | **93.53** | **92.44** |
| QWEN3-8B | DIRECT | 82.71 | 69.32 | 81.46 | 82.14 | 76.79 | 81.65 | 86.46 | 79.71 | 85.00 |
| | CoT | 82.94 | 71.09 | 81.33 | 83.18 | 75.95 | 82.95 | 86.54 | 81.88 | 86.59 |
| | **OURS** | **87.76** | **94.89** | **92.01** | **87.55** | **93.42** | **90.73** | **89.16** | **93.94** | **91.18** |

*Table 4.* Necessity and sufficiency of RAG-retrieved statements.

| NECESSITY | SUFFICIENCY | GPT-4.1 | HUMAN |
|-----------|-------------|---------|-------|
| ✓ | ✓ | 4.37% | 2.00% |
| ✓ | ✗ | 17.65% | 24.00% |
| ✗ | ✓ | 9.42% | 11.00% |
| ✗ | ✗ | 68.56% | 63.00% |

*Table 5.* Ablation studies on translation, completion, and verifiers.

| TRANS. | COMP. | LOGICVER | FACTVER | ACC. | RECALL | F1 |
|--------|-------|----------|---------|------|--------|-----|
| | | | | 50.64 | 42.78 | 58.78 |
| ✓ | | ✓ | | 51.78 | 46.78 | 61.49 |
| | ✓ | | ✓ | 55.71 | 52.20 | 66.39 |
| ✓ | ✓ | | ✓ | 59.04 | 58.96 | 68.78 |
| ✓ | ✓ | ✓ | | 64.27 | 65.86 | 73.25 |
| ✓ | ✓ | ✓ | ✓ | **68.53** | **69.39** | **78.39** |

gains on Logic and RAG Baselines. Symbolic solvers (e.g., LogicLM, LINC) struggle in open-world scenarios because they lack the necessary premises to complete a formal proof, often returning "Unknown" or "False". RAG-based approaches fail to provide the precise logical links required, as they rely on statistical similarity rather than logical necessity. In the HyDE+Ours and SiReRAG+Ours baselines, we add an RAG stage before OpenIKLR to extract relevant knowledge from an external database ATOMIC2020 (Hwang et al., 2021), which is then fed into the original contexts.

The experiments demonstrate that OpenIKLR outperforms consistently RAG with OpenIKLR, indicating that involving unnecessary premises through RAG can introduce noise that interferes with the reasoning process generated by OpenIKLR alone. In sum, the experimental results establish a performance hierarchy: RAG, CoT, RAG+OpenIKLR,

OpenIKLR, indicating that RAG often recalls insufficient or unnecessary premises that achieve marginal improvements over CoT while introducing noise that adversely affects the precision of OpenIKLR. To evaluate the quality of the statements retrieved by RAG, we used GPT-4.1 to assess the entire dataset and additionally recruited human experts to annotate 100 randomly sampled cases. Table 4 shows that most retrievals are neither necessary nor sufficient, which explains the poor performance of RAG methods. More experiments on different RAG settings are in Appendix E.1.

### 4.3. RQ 2: Ablation Study

We conducted ablation experiments using DeepSeek-V3.2 on Multi-LogiEval-90%. Table 5 demonstrates that every module, including the Translator, Completion module, Logic Verifier, and Fact Verifier, within the OpenIKLR framework contributes significantly to its reasoning performance. Starting from a baseline F1 score of 58.78%, the addition of the Translator and Logic Verifier provides an improvement to 61.49%, confirming that symbolic translation and reasoning are helpful even without external knowledge. The Completion module brings more substantial performance gains, underscoring the critical necessity of identifying and filling logical gaps in open-world scenarios. Furthermore, the results highlight the indispensable role of our dual-verification mechanism. Comparing the configuration in Row 5 with the full pipeline in Row 6 shows that the Fact Verifier provides a final crucial boost, increasing the F1 score from 73.25% to 78.39%. Similarly, Row 4 to Row 6 verify the importance of Logic Verifier, showing nearly 10% F1 score improvement.

*Table 6.* Performance comparison w/ and w/o minimal constraints.

| MODEL | CONSTRAINT | ACC.↑ | AVG. NUM.↓ | FACT ACC.↑ |
|---|---|---|---|---|
| DEEPSEEK-V3.2 | w/o | 64.52 | 6.18 | 98.58 |
| | **W/** | **68.53** | **1.78** | **99.19** |
| GPT-5 | w/o | 60.86 | 8.98 | **98.81** |
| | **W/** | **66.95** | **1.66** | 97.60 |
| QWEN3-235B-A22B | w/o | 62.57 | 10.94 | 96.09 |
| | **W/** | **69.53** | **2.47** | **96.53** |

*Table 7.* Cost comparison under minimal constraints and maximal constraints.

| CONSTRAINT | TOKEN NUM↓ | API CALLS↓ | ACC. | RECALL | F1 |
|---|---|---|---|---|---|
| MAXIMAL | 3572.94 | 5.31 | 85.28 | 83.73 | 86.15 |
| MINIMAL | **3106.92** | **4.77** | **90.81** | **94.12** | **91.82** |

*Table 8.* Step-wise efficiency of OpenIKLR.

| TRANS. | LOGICVER. | FACTVER. | LABEL | TOKEN NUM↓ | API CALLS↓ |
|---|---|---|---|---|---|
| ✗ | N/A | N/A | ALL | 811.26 | 2.00 |
| ✓ | ✗ | N/A | T | 3401.82 | 5.00 |
| ✓ | ✗ | N/A | F | 3573.68 | 5.00 |
| ✓ | ✓ | ✗ | T | 2455.07 | 6.23 |
| ✓ | ✓ | ✗ | F | 2909.23 | 5.76 |
| ✓ | ✓ | ✓ | T | 1358.50 | 2.49 |
| ✓ | ✓ | ✓ | F | 2131.48 | 4.16 |

### 4.4. RQ 3: Sensitivity of Hyperparameters

As shown in Figure 4, we study the impact of the maximum number of completion iterations, $K_{max}$, on the performance of OpenIKLR across various backbones. The results demonstrate a clear upward trend in F1 scores as $K_{max}$ increases, indicating that additional iterations allow the model to more effectively complete the missing premises. Notably, the performance growth becomes marginal once $K_{max}$ reaches 4, with most models showing a plateau in performance gains. Consequently, to achieve an optimal trade-off between effectiveness and computational efficiency, we select $K_{max} = 4$ as the standard hyperparameter for our primary experiments.

### 4.5. In-depth Analysis

**Solver Verification Dynamics.** We further analyze the solver verification dynamics across different backbones and iteration counts. As shown in Figure 5, the solver verification pass rate is largest at $k = 1$, where it captures nearly 30% of the dataset across different models, before gradually decreasing in later iterations. The F1 scores for these verified cases remain consistently high—typically above 90% across all values of $k$. This trend demonstrates that OpenIKLR's dual-verification mechanism effectively ensures that the generated premises are both logically sufficient and factually accurate, maintaining rigorous reasoning quality throughout the iterative process.

**The Effectiveness of the Minimal Constraints.** To evaluate the effectiveness of the minimal premise set preference in our method, we conduct an experiment comparing the

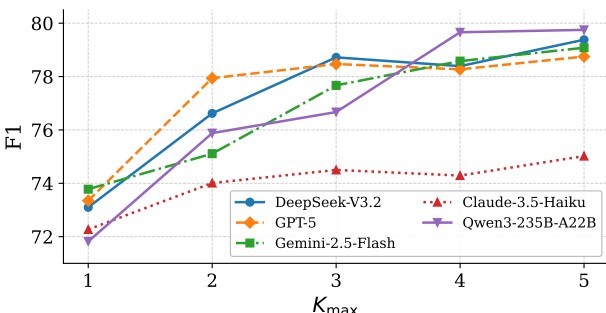

*Figure 4.* F1 scores of our method on the Multi-LogiEval-90% using different $K_{max}$.

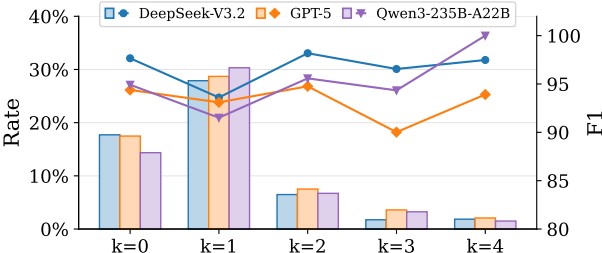

*Figure 5.* Solver verification pass rate after the $k$-th completion (bar), and F1 score of verified cases (line) on Multi-LogiEval-90%.

performance of OpenIKLR w/ and w/o the minimal premise set constraint in the prompt. Table 6 presents the experimental results on Multi-LogiEval-90%, showing that with the minimal premise set constraint, we can achieve higher reasoning accuracy. Meanwhile, we find that the average number of premises added drops significantly. The factual accuracy is relatively stable, indicating that the LLMs are capable of generating high-quality factual knowledge. All results can be found in the Appendix E.3.

We further conduct a stricter comparison on FOLIO-90% by replacing the minimal constraint prompt with a maximal constraint prompt. As shown in Table 7, the minimal constraint reduces the average token cost from 3572.94 to 3106.92 and the average number of API calls from 5.31 to 4.77, while also improving overall performance. In addition, we find the maximal constraint increases the average time by 59.57% compared with the minimal constraint. These results suggest that the minimal constraint is not merely an efficiency heuristic but also helps reduce noise in the added premises and improve downstream logical reasoning.

**Efficiency Analysis.** We decompose the computational cost of OpenIKLR according to the algorithm step in Table 8. In brief, our algorithm first translates natural language into first-order logic and invokes the solver. When the proof attempt fails, OpenIKLR triggers up to $K_{max}$ minimal iterations guided by the failed trace. Once logical verification succeeds, the completed premises are further checked by the

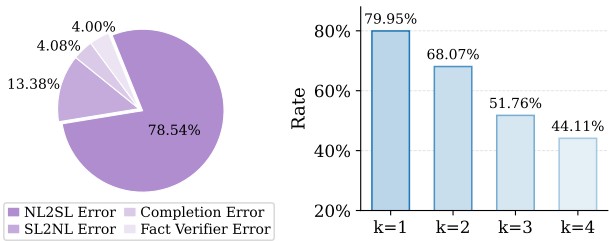

*Figure 6.* Left: OpenIKLR failure mode categorization. Right: Frequency of logical success but factual failure.

*Table 9.* Performance on two real-world legal datasets.

| MODEL | METHOD | ACC. | RECALL | F1 |
|---|---|---|---|---|
| **CAIL2024** | | | | |
| DEEPSEEK-V3.2 | CoT | 79.27 | 96.26 | 84.63 |
| | **OURS** | **85.57** | **98.05** | **86.28** |
| GPT-5 | CoT | 77.51 | 86.37 | 78.64 |
| | **OURS** | **80.85** | **90.86** | **82.29** |
| QWEN3-235B-A22B | CoT | 80.65 | 95.36 | 82.83 |
| | **OURS** | **83.54** | **95.78** | **85.07** |
| **LEGALBENCH** | | | | |
| DEEPSEEK-V3.2 | CoT | 56.99 | 43.38 | 50.21 |
| | **OURS** | **59.36** | **60.23** | **58.17** |
| GPT-5 | CoT | 63.60 | 82.21 | 73.92 |
| | **OURS** | **66.72** | **83.58** | **76.37** |
| QWEN3-235B-A22B | CoT | 58.67 | 39.71 | 49.09 |
| | **OURS** | **61.75** | **54.69** | **58.21** |

fact verifier. The relative inefficiency mainly arises when logical verification succeeds but fact verification fails, because these cases have already paid for solver-guided completion and verifier calls. This effect is especially visible for false samples, which require 2909.23 tokens and 5.76 API calls in the logic-pass but fact-fail branch. When both logic and fact verification pass, true samples are resolved more directly, while false samples still require more iterations and verification calls on average.

**Failure Mode Categorization.** We manually inspect the failure cases and categorize each case according to the first stage at which the pipeline fails. Specifically, NL2SL Error refers to errors in translating natural language into symbolic language; Completion Error refers to incorrect premise completion; SL2NL Error refers to errors in translating completed symbolic premises back into natural language; and Fact Verifier Error refers to cases where the fact verifier incorrectly judges the validity of the premise. As shown in Figure 6 (left), the dominant source of failure is NL2SL Error, which accounts for 78.54% of all inspected failures.

This suggests that the main remaining bottleneck of OpenIKLR is accurate symbolic parsing rather than the premise completion or verification modules. We further analyze the frequency with which logical verification succeeds while factual verification fails. As shown in Figure 6 (right), 79.95% of logically verified samples are rejected by the fact verifier in the first round. This rate decreases from 79.95% at $k = 1$ to 44.11% at $k = 4$, but remains substantial across iterations. This pattern shows that symbolic sufficiency alone is not enough. The solver can repeatedly certify logically useful premises that are not factually reliable, thereby justifying the need for fact verification.

## 5. Real-World Application

To further evaluate OpenIKLR beyond logical reasoning benchmarks, we conduct real-world experiments in the legal domain. We evaluate on two legal benchmarks: CAIL2024[1] and LegalBench (Guha et al., 2023). CAIL2024 is a Chinese legal benchmark, from which we randomly sample 300 cases. LegalBench is an English legal reasoning bench-

mark centered on U.S. law. We select the SARA Entailment task, retain the original "description" and "question" fields, and formulate the evaluation as a legal QA task. We adapt original prompts to fit the legal domain, requiring the framework to supplement relevant legal knowledge or articles and specific statutes while verifying that all generated premises align with authentic legal articles. As shown in Table 9, OpenIKLR outperforms the baseline across all models. On LegalBench, OpenIKLR improves F1 score by 7.96%, 2.45%, and 9.12% over CoT for DeepSeek-V3.2, GPT-5, and Qwen3-235B-A22B, respectively. Similar gains are observed on CAIL2024, where our framework bridges reasoning gaps by completing implicit legal knowledge. These results show OpenIKLR is not only robust for general logic benchmarks, but also effective in specialized, knowledge-intensive domains like legal judgment prediction.

## 6. Conclusion

In this paper, we proposed OpenIKLR, an Open-world Incomplete-Knowledge-aware Logical Reasoning framework designed to bridge the "reasoning gap" in scenarios where provided information is insufficient to logically deduce a given conclusion. By integrating the formal logical reasoning via symbolic solvers and the semantic reasoning via world knowledge of LLMs, our approach addresses the limitations of the closed-world assumption by identifying missing premises or implicit commonsense knowledge. Through an iterative completion approach with a dual-verification process, OpenIKLR ensures that any added knowledge is both logically sound and factually accurate. Extensive experiments demonstrate that OpenIKLR stably outperforms 11 state-of-the-art logical reasoning and RAG-based baselines across real-world datasets, providing new insights for handling incomplete information in complex open-world logical reasoning. A current limitation is that our evaluation focuses on first-order logic, leaving extensions to richer logical formalisms for future investigation.

---

[1] https://github.com/china-ai-law-challenge/CAIL2024

## Acknowledgments

This work is supported by the Beijing Natural Science Foundation (No. L257007), the NSF China (No. 62476245, No. 62276004), the Beijing Major Science and Technology Project (No. Z251100008425006, No. Z251100008125054), the UKRI grant: Turing AI Fellowship EP/W002981/1, the State Key Laboratory of General Artificial Intelligence, Tsinghua University's Initiative for Advancing First-Class and World-Leading Disciplines in the Humanities and Social Sciences, the Beijing Academy of Artificial Intelligence (BAAI), and the Microsoft Ltd.

## Impact Statement

By integrating symbolic logic solvers with large language models, OpenIKLR enables more robust and interpretable reasoning when required premises are missing or implicit, which reflects real-world decision-making scenarios more closely than closed-world assumptions. The primary positive impact of this work lies in enhancing the reliability of reasoning systems in safety-critical and knowledge-intensive domains such as legal analysis, scientific reasoning, education, and decision support. By explicitly identifying and verifying missing premises, the framework reduces reliance on spurious correlations and uncontrolled retrieval, contributing to more transparent and verifiable model behavior. Potential risks include misuse in domains where incorrect or biased background knowledge may still be generated or verified imperfectly.

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

# Appendix

# A. Related Work

## A.1. Logical Reasoning Methods in LLMs

Many previous studies have shown the possibility and necessity of introducing neural symbolic techniques to improve the reasoning capability of LLMs (Jia et al., 2025; Shao et al., 2025; Yang et al., 2026b). Among them, logical reasoning in LLMs is one of the primary research directions (Cheng et al., 2025; Zhou et al., 2026), with methodologies generally falling into three paradigms: external solver-based approach, prompting-based approach, and fine-tuning approach.

Solver-based frameworks to enhance the logical reasoning capabilities of LLMs mitigate the inherent probabilistic failures of LLMs by leveraging external logic solvers to perform the reasoning process. These methods usually utilize the LLM to translate logical question-answering problems in natural language into formal symbolic representations. A logic solver then executes the reasoning over these symbols, often utilizing ensemble techniques to get the final answer. Specifically, SatLM (Ye et al., 2023) utilizes declarative prompting to combine the linguistic capabilities of LLMs with the rigorous verification of SAT solvers. LINC (Olausson et al., 2023) translates natural language problems into several symbolic representations candidates to enable faithful logical reasoning through external solvers. In contrast to LINC and SatLM, which rely on a single symbolic logic formalism and its associated solver, LogicLM (Pan et al., 2023) employs LLMs to map natural language inputs into dataset-dependent, task-specific representations, such as linear programming, first-order logic, constraint satisfaction problems, or SAT. The translated representations are then solved using the corresponding symbolic solvers. To improve the accuracy of symbolic parsing, CLOVER (Ryu et al., 2025) decomposes an input natural language passage into fine-grained clauses and explicitly models their dependency relations, before mapping these units into a symbolic representation. VERUS-LM (Callewaert et al., 2025) incorporates an iterative correction mechanism, where signals from a downstream reasoning module are leveraged to revise flawed logical forms at both the structural and meaning levels. ∀uto∃val (Karia et al., 2024) further proposes a cyclic translation framework that alternates between symbolic expressions and natural language paraphrases, and employs automated solver-based equivalence checking to validate consistency across translation rounds, thereby reducing reliance on manual supervision.

Approaches based on prompting can be broadly divided into two categories. One line of work encourages models to externalize intermediate reasoning structures while answering queries, for example, by organizing inference into chains, trees, or diagrammatic forms (Wei et al., 2022; Yao et al., 2023a; Zhang et al., 2024b). Another line adopts a staged prompting pipeline, in which the language model is guided through a sequence of subtasks, including formalizing the input, designing plans, carrying out incremental inference, and checking the final outcome. ChatLogic (Wang et al., 2024) further embeds symbolic logic problems within interactive prompting to mitigate syntactic and semantic inconsistencies. SymbCoT (Xu et al., 2024b) guides models to convert NL inputs into SL, apply explicit inference rules, and subsequently validate both the formalization and the derived conclusions. LoT (Liu et al., 2025d) enriches prompts with rule-based expansions of implicit logical content.

Deficiencies in LLM reasoning performance are often linked to the scarcity of explicit, high-quality logical derivations—particularly multi-step proofs—within pretraining data (Morishita et al., 2024). To mitigate this issue, fine-tuning methods typically rely on automatically generated corpora or curated collections of intermediate reasoning traces, enabling models to better learn structured inference while improving transparency. Representative efforts include logic-aware contrastive objectives (Wang et al., 2022). For enhanced interpretability, LOGIPT (Feng et al., 2024) aligns natural language inputs with solver-style symbolic steps, while ALT (Morishita et al., 2024) synthesizes deduction sequences for stepwise supervision. LogicAsker (Wan et al., 2024) further adopts adaptive fine-tuning driven by formal-logic skills and error diagnosis, and LogicLLM (Jiao et al., 2024) reduces human labeling through a fully self-supervised training paradigm.

While the previous methods operate under a closed-world assumption, where presuming all necessary information is provided in the context. OpenIKLR addresses the open-world setting where information is often incomplete. Unlike standard solver-based systems that fail when premises are missing, our framework identifies "reasoning gaps" through a symbolic solver. It then employs an iterative completion mechanism to generate the minimal set of missing premises.

## A.2. Knowledge Completion Methods in LLMs Reasoning

LLMs have demonstrated strong performance on a wide range of reasoning tasks. However, their reasoning capability is still constrained by the limited coverage of knowledge implicitly stored in model parameters. To address these gaps during

reasoning in dynamic open-world (Liu et al., 2025c), a growing body of work has explored various knowledge completion and augmentation methods (Liu et al., 2024), which can be broadly categorized into parameterized knowledge enhancement, retrieval-augmented generation (RAG), and explicit knowledge completion during the reasoning process.

Parameterized knowledge enhancement methods aim to inject external knowledge into model parameters during pretraining or fine-tuning, enabling models to implicitly access such knowledge at inference time. Early studies incorporate structured knowledge, such as knowledge graph triples or symbolic facts, into language model pretraining to improve factual recall and reasoning ability (Peters et al., 2019; Sun et al., 2019). Subsequent work adopts continual pretraining or large-scale instruction tuning on knowledge-intensive datasets to further enhance reasoning performance (Wang et al., 2021; Chowdhery et al., 2023). Despite their effectiveness, these approaches encode knowledge in a static manner, making it difficult to adapt to dynamically evolving.

RAG has become a dominant paradigm for knowledge completion during inference. These methods retrieve relevant information from external knowledge sources, such as document collections, Wikipedia, or search engines, and incorporate the retrieved evidence into the model's input. Representative approaches, including REALM (Guu et al., 2020), RAG (Lewis et al., 2020), and FiD (Hofstätter et al., 2023), demonstrate substantial improvements on knowledge-intensive reasoning tasks. Follow-up work explores tighter integration between retrieval and reasoning, such as multi-hop and iterative retrieval (Yang et al., 2018; Trivedi et al., 2023), retrieval result re-ranking (Mortaheb et al., 2025), and dynamically triggering retrieval at different stages of the reasoning process (Liu et al., 2025b). However, the performance of RAG heavily depends on retrieval quality and may introduce redundant or noisy information in complex reasoning settings.

Explicit knowledge completion during reasoning focuses on augmenting missing knowledge within intermediate reasoning steps. CoT prompting (Wei et al., 2022) explicitly elicits intermediate reasoning paths, allowing models to introduce implicit premises and commonsense knowledge, thereby improving performance on complex reasoning tasks. Based on this, subsequent work proposes self-questioning mechanisms (Press et al., 2023), knowledge prompting strategies (Cui et al., 2024), and reasoning frameworks that leverage programs or external tools (Schick et al., 2023; Yao et al., 2023b), enabling models to actively identify and supplement missing knowledge during inference. Compared to single-step retrieval-based methods, these approaches more tightly couple knowledge completion with the reasoning structure.

Unlike current methods, OpenIKLR uses a logic solver to precisely identify missing premises and iteratively generates a minimal set of necessary premises with LLMs, incorporating dual logic and fact verification to ensure both logical sufficiency and factual correctness, while preserving training-free flexibility and interpretable intermediate reasoning steps.

## B. Algorithmic Description

Algorithm 1 presents the specific details of OpenIKLR. OpenIKLR integrates symbolic reasoning with LLMs. It first translates natural language premises and the conclusion into an executable symbolic representation and performs direct verification using a logical solver. If the verification succeeds, OpenIKLR returns a prediction (True). Otherwise, the algorithm enters an iterative process with a fixed maximum number of iterations, where in each round the model supplements potentially missing implicit premises based on feedback from the failed reasoning attempt and re-invokes the solver for verification. This iteration limit is used to control reasoning cost and prevent unbounded completion. When the augmented premises are sufficient to support the conclusion, OpenIKLR does not accept the result immediately. Instead, it further applies a fact verification step to the new premises and returns a prediction only if this verification succeeds. Repeated symbolic verification suggests that the conclusion holds but no reliable factual support can be obtained, then OpenIKLR outputs a prediction (False). In cases where symbolic reasoning remains unstable or fails to converge, the model falls back to a CoT-based prediction as the final output.

## C. Theoretical Analysis

### C.1. Proof of Theorem 3.1

*Proof.* We provide a complete proof based on standard PAC-Bayes techniques. For any measurable function $g(\Delta)$, the Donsker–Varadhan change of measure inequality gives:

$$\mathbb{E}_{\Delta \sim Q}[g(\Delta)] \ \leq \ \mathrm{KL}(Q\|P) + \log \mathbb{E}_{\Delta \sim P}[e^{g(\Delta)}].$$

---

**Algorithm 1** OpenIKLR Algorithm

---

**Input**: A set of natural language premises $\mathcal{P}$ and a natural language conclusion $\mathcal{C}$
**Output**: Predicted label $\mathcal{Y}$

1:  $(\mathcal{P}_s, \mathcal{C}_s) \leftarrow \text{TRANSLATOR}(\mathcal{P}, \mathcal{C})$          ▷ *Translate natural language into symbolic language*
2:  $(\mathcal{V}, \mathcal{R}, \mathcal{T}) \leftarrow \text{SOLVER}(\mathcal{P}_s, \mathcal{C}_s)$     ▷ *Solver verifies logic and outputs execution status $\mathcal{V}$, result $\mathcal{R}$, and proof $\mathcal{T}$*
3:  $k \leftarrow 0, n \leftarrow 0$          ▷ *Initialize iteration count $k$ and the true $\mathcal{R}$ count $n$*
4:  **if** $\mathcal{V}$ is successful **and** $\mathcal{R}$ is True **then**
5:    **return** True
6:  **else**
7:    **while** $k < K_{\max}$ **do**
8:     $\mathcal{P}_s' \leftarrow \text{COMPLETION}(\mathcal{P}, \mathcal{C}, \mathcal{P}_s, \mathcal{C}_s, \mathcal{T})$       ▷ *LLM generates new symbolic premises $\mathcal{P}_s'$*
9:     $(\mathcal{V}, \mathcal{R}, \mathcal{T}) \leftarrow \text{SOLVER}(\mathcal{P}_s \cup \mathcal{P}_s', \mathcal{C}_s)$
10:    **if** $\mathcal{V}$ is successful **and** $\mathcal{R}$ is True **then**
11:     $n \leftarrow n + 1$
12:     $\mathcal{P}' \leftarrow \text{TRANSLATOR}(\mathcal{P}_s')$       ▷ *Translate symbolic language into natural language*
13:     $\mathcal{F} \leftarrow \text{FACTVERIFIER}(\mathcal{P}, \mathcal{C}, \mathcal{P}', \mathcal{P}_s')$      ▷ *LLM verifies if new premises are fact*
14:     **if** $\mathcal{F}$ is True **then**
15:      **return** True
16:     **end if**
17:    **end if**
18:    $k \leftarrow k + 1$
19:   **end while**
20:   **if** $\mathcal{V}$ is successful **and** $n > K_{\max}/2$ **then**
21:    **return** False
22:   **else**
23:    **return** $\mathcal{Y}_{CoT}$             ▷ *Use label generated by CoT*
24:   **end if**
25: **end if**

---

Fix $\lambda > 0$ and define

$$g(\Delta) = \lambda\big(R_{\mathcal{D}}(f_\Delta) - \hat{R}_S(f_\Delta)\big).$$

By Hoeffding's inequality, for any fixed $\Delta$,

$$\mathbb{E}_{S \sim \mathcal{D}^n}\big[e^{\lambda(R_{\mathcal{D}}(f_\Delta) - \hat{R}_S(f_\Delta))}\big] \leq e^{\frac{\lambda^2}{2(n-1)}}.$$

Taking expectation over $\Delta \sim P$ yields:

$$\mathbb{E}_{\Delta \sim P}\mathbb{E}_S\big[e^{g(\Delta)}\big] \leq e^{\frac{\lambda^2}{2(n-1)}}.$$

By Markov's inequality, with probability at least $1 - \delta$ over $S$,

$$\mathbb{E}_{\Delta \sim P}\big[e^{g(\Delta)}\big] \leq \frac{1}{\delta}e^{\frac{\lambda^2}{2(n-1)}}.$$

Substituting into the change of measure inequality gives, with probability at least $1 - \delta$:

$$\lambda\big(R_{\mathcal{D}}(Q) - \hat{R}_S(Q)\big) \leq \text{KL}(Q\|P) + \log\frac{1}{\delta} + \frac{\lambda^2}{2(n-1)}.$$

Rearranging,

$$R_{\mathcal{D}}(Q) \leq \hat{R}_S(Q) + \frac{\text{KL}(Q\|P) + \log(1/\delta)}{\lambda} + \frac{\lambda}{2(n-1)}.$$

By choosing

$$\lambda = \sqrt{2(n-1)(KL(Q\|P) + \log(1/\delta))},$$

we obtain:

$$R_{\mathcal{D}}(Q) \leq \hat{R}_S(Q) + \sqrt{\frac{\text{KL}(Q\|P) + \log(1/\delta)}{2(n-1)}}.$$

This completes the proof.                   $\square$

## C.2. Lemma

**Lemma C.1** (KL Decomposition). *Let $Z = \sum_{\Delta \subseteq \mathcal{P}} e^{-\alpha|\Delta|}$ for the prior $P(\Delta) \propto \exp(-\alpha|\Delta|)$, the KL divergence decomposes as*

$$\mathrm{KL}(Q\|P) = \alpha\, \mathbb{E}_Q[|\Delta|] + \mathbb{E}_Q[\log Q(\Delta)] + \log Z.$$

*Proof.* By definition,

$$\mathrm{KL}(Q\|P) = \mathbb{E}_Q[\log Q(\Delta)] - \mathbb{E}_Q[\log P(\Delta)].$$

Substituting $\log P(\Delta) = -\alpha|\Delta| - \log Z$ yields the result. $\qquad\square$

## C.3. Proof of Theorem 3.2

*Proof.* By Lemma C.1,

$$\mathrm{KL}(Q\|P) = \alpha\, \mathbb{E}_Q[|\Delta|] + \mathbb{E}_Q[\log Q(\Delta)] + \log Z.$$

The normalization constant $\log Z$ is independent of $Q$. Thus, comparing $Q_1$ and $Q_2$,

$$\mathrm{KL}(Q_2\|P) - \mathrm{KL}(Q_1\|P) = \alpha\big(\mathbb{E}_{Q_2}[|\Delta|] - \mathbb{E}_{Q_1}[|\Delta|]\big) + \big(\mathbb{E}_{Q_2}[\log Q_2(\Delta)] - \mathbb{E}_{Q_1}[\log Q_1(\Delta)]\big).$$

By assumption, both terms on the right-hand side are nonnegative, and the first term is strictly positive. Hence

$$\mathrm{KL}(Q_2\|P) > \mathrm{KL}(Q_1\|P).$$

Since the PAC-Bayes bound in Theorem 3.1 is a strictly increasing function of $\mathrm{KL}(Q\|P)$, the result follows. $\qquad\square$

Under a cardinality-aware prior, we prove that the PAC-Bayes generalization bound is an increasing function of the expected number of completed premises. This establishes a monotonic relationship between premise completion and generalization error that is independent of the specific solver or translation module.

# D. Implementation Details

**Dataset Quality Control.** During dataset construction, we use GPT-4.1 with the temperature set to 1.5 and run the classification process six times for each premise. A premise is considered real-world knowledge or common sense only if it is identified as such in at least five out of six runs, and only these premises are eligible for masking. To further verify that our dataset construction aligns with real-world conditions, we conduct an agreement analysis with multiple LLMs and human annotators. Specifically, Claude-4.5, Doubao-2, Gemini-3, GPT-4.1, Qwen-Max, and human annotators independently judge whether each candidate premise expresses real-world or commonsense knowledge. As shown in Figure 7, the resulting labels are highly consistent across evaluators and almost all pairwise agreements exceed 92%. This high agreement indicates that our premise identification aligns well with human judgment, supporting the reliability of the dataset construction.

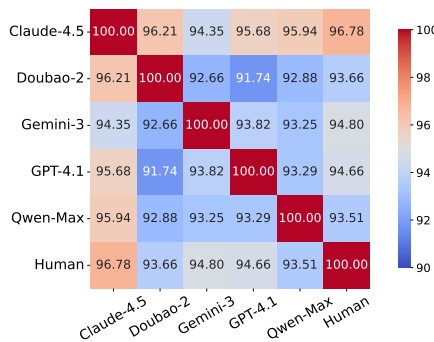

*Figure 7.* Agreement across LLMs and human.

**Other Details.** To control the difficulty of the open-world setting, we further limit the maximum number of masked premises per sample to four. For the FOLIO dataset, 35.0% of samples have one premise masked, 39.0% have two premises masked, 22.2% have three premises masked, and 3.8% have four premises masked. For the Multi-LogiEval dataset, 62.9% of samples have one premise masked, 25.6% have two premises masked, 10.8% have three premises masked, and 0.7% have four premises masked. During evaluation, we set the temperature of GPT-5 to 1, as this is the only supported setting. For all other models, the temperature is set to 0. For RAG-based methods, we adopt Qwen3-8B-Embedding as the embedding model, and we use ATOMIC2020 as the external knowledge base. We set the retrieval Top-k to 3 for each query to avoid introducing excessive or weakly relevant external knowledge.

*Table 10.* Effect of different RAG configurations on FOLIO-90%. The first two columns indicate which configuration is changed and the corresponding setting.

| Configuration | Setting | Method | Acc. | Recall | F1 |
|---|---|---|---|---|---|
| Knowledge Base | ATOMIC2020 | RAG | 82.26 | 71.68 | 81.54 |
| | | RAG+**Ours** | 86.59 | 85.84 | 87.52 |
| | Wikipedia.en | RAG | 81.74 | 69.62 | 80.68 |
| | | RAG+**Ours** | 87.24 | 88.20 | 88.33 |
| | ConceptNet | RAG | 78.68 | 63.72 | 76.60 |
| | | RAG+**Ours** | 84.65 | 79.06 | 84.94 |
| Top-$k$ | 3 | RAG | 82.26 | 71.68 | 81.54 |
| | | RAG+**Ours** | 86.59 | 85.84 | 87.52 |
| | 5 | RAG | 83.20 | 72.86 | 82.61 |
| | | RAG+**Ours** | 86.95 | 84.37 | 87.80 |
| | 10 | RAG | 81.88 | 70.86 | 80.44 |
| | | RAG+**Ours** | 85.24 | 83.37 | 85.86 |
| | 15 | RAG | 81.84 | 69.94 | 80.33 |
| | | RAG+**Ours** | 85.24 | 83.02 | 85.19 |
| | 20 | RAG | 81.04 | 68.39 | 80.17 |
| | | RAG+**Ours** | 85.08 | 83.02 | 85.06 |
| Embed Model | Qwen3-8B-Embedding | RAG | 82.26 | 71.68 | 81.54 |
| | | RAG+**Ours** | 86.59 | 85.84 | 87.52 |
| | llama-embed-nemotron-8b | RAG | 80.91 | 69.39 | 80.21 |
| | | RAG+**Ours** | 86.91 | 87.32 | 87.96 |
| | BGE-M3 | RAG | 80.26 | 66.73 | 79.07 |
| | | RAG+**Ours** | 84.49 | 81.42 | 85.19 |
| | NV-Embed-v2 | RAG | 79.45 | 64.37 | 77.81 |
| | | RAG+**Ours** | 86.27 | 83.78 | 86.98 |
| Reranker | None | RAG | 82.26 | 71.68 | 81.54 |
| | | RAG+**Ours** | 86.59 | 85.84 | 87.52 |
| | Qwen3-Reranker-8B | RAG | 83.49 | 75.29 | 83.52 |
| | | RAG+**Ours** | 86.91 | 86.43 | 87.86 |
| | BGE-Reranker-large | RAG | 85.11 | 77.35 | 85.22 |
| | | RAG+**Ours** | 87.10 | 85.88 | 87.95 |
| N/A | N/A | **Ours** | **90.81** | **94.12** | **91.82** |

## E. Detailed Experiments Results

### E.1. Results of Different RAG Configurations

We further analyze whether the performance gap between RAG augmentation and OpenIKLR can be closed by changing the retrieval configuration. All experiments in this subsection are conducted on FOLIO-90%. The base setting uses SiReRAG, ATOMIC2020, Top-$k$ = 3, Qwen3-8B-Embedding, and no reranker.

Table 10 summarizes the effects of four configurations: knowledge base, retrieval Top-$k$, embedding model, and reranker. Across ATOMIC2020, Wikipedia.en, and ConceptNet, adding our framework consistently improves the RAG baseline. Changing Top-$k$ or the embedding model yields only limited variation. Adding a reranker improves RAG, especially in recall, but the best reranked RAG result is still substantially lower than OpenIKLR. These results indicate that stronger retrieval and ranking help, but do not remove the fundamental mismatch between retrieved evidence and the minimal missing premises.

## E.2. Results of Qwen3-235B-A22B

As a state-of-the-art model in the Qwen3 series, we also evaluated our logic reasoning method and RAG-based method on Qwen3-235B-A22B. As shown in Table 11, OpenIKLR achieves a substantial performance improvement.

*Table 11.* Comparison of Qwen3-235B-A22B on **Multi-LogiEval** and **FOLIO**. The percentages indicate the proportion of open-world samples in each subset. **Bold** indicates the best performance within each model.

| MODEL | METHOD | 90% | | | 70% | | | 50% | | |
|---|---|---|---|---|---|---|---|---|---|---|
| | | ACC. | RECALL | F1 | ACC. | RECALL | F1 | ACC. | RECALL | F1 |
| | | **MULTI-LOGIEVAL** | | | | | | | | |
| | DIRECT | 50.50 | 42.09 | 58.31 | 58.23 | 49.64 | 65.16 | 61.00 | 53.51 | 69.08 |
| | COT | 52.07 | 45.04 | 60.73 | 57.37 | 50.00 | 64.86 | 63.00 | 57.89 | 71.82 |
| | LINC | 17.31 | 19.65 | 28.11 | 18.31 | 20.91 | 28.71 | 23.29 | 26.32 | 35.84 |
| | LOGICLM | 31.47 | 31.48 | 43.04 | 33.91 | 34.91 | 45.39 | 42.43 | 45.61 | 56.34 |
| | CR | 36.08 | 24.45 | 38.61 | 43.09 | 29.92 | 45.25 | 49.34 | 39.92 | 56.16 |
| QWEN3-235B-A22B | DETERMLR | 33.05 | 20.17 | 33.14 | 36.34 | 20.73 | 33.88 | 41.71 | 30.18 | 45.74 |
| | SYMBCOT | 48.98 | 43.33 | 58.10 | 56.06 | 50.48 | 64.63 | 52.21 | 50.47 | 62.43 |
| | HYDE | 36.20 | 21.31 | 34.27 | 43.49 | 27.87 | 42.54 | 44.60 | 31.67 | 47.39 |
| | HYDE+**OURS** | 53.20 | 52.93 | 63.84 | 55.78 | 56.23 | 65.62 | 55.99 | 57.36 | 67.25 |
| | SIRERAG | 26.61 | 11.30 | 20.22 | 34.91 | 18.15 | 30.53 | 37.57 | 24.69 | 39.22 |
| | SIRERAG+**OURS** | 54.47 | 57.04 | 67.29 | 55.95 | 58.53 | 67.73 | 55.57 | 58.66 | 68.26 |
| | **OURS** | **69.53** | **72.52** | **79.66** | **69.67** | **74.18** | **79.38** | **73.57** | **79.12** | **82.98** |
| | | **FOLIO** | | | | | | | | |
| | DIRECT | 80.61 | 66.08 | 78.87 | 78.55 | 74.26 | 78.81 | 84.54 | 76.81 | 81.83 |
| | COT | 76.09 | 69.03 | 75.97 | 78.41 | 74.26 | 78.75 | 82.69 | 78.99 | 82.89 |
| | LINC | 32.15 | 39.23 | 38.78 | 41.82 | 51.48 | 48.80 | 51.92 | 63.04 | 58.19 |
| | LOGICLM | 37.80 | 44.25 | 43.80 | 46.82 | 55.27 | 52.82 | 57.31 | 67.39 | 62.63 |
| | CR | 64.52 | 67.50 | 70.41 | 67.05 | 67.64 | 76.59 | 78.08 | 71.16 | 77.84 |
| QWEN3-235B-A22B | DETERMLR | 75.81 | 63.86 | 75.01 | 77.59 | 68.22 | 77.12 | 80.23 | 72.16 | 79.89 |
| | SYMBCOT | 63.86 | 47.22 | 60.18 | 64.77 | 49.51 | 62.20 | 74.70 | 63.04 | 73.42 |
| | HYDE | 74.65 | 66.40 | 74.50 | 76.59 | 69.75 | 76.50 | 80.00 | 76.96 | 80.23 |
| | HYDE+**OURS** | 82.39 | 84.96 | 84.09 | 81.59 | 87.34 | 83.64 | 85.77 | 94.93 | 87.63 |
| | SIRERAG | 69.63 | 49.85 | 64.26 | 74.55 | 57.81 | 70.98 | 77.31 | 64.49 | 75.11 |
| | SIRERAG+**OURS** | 77.76 | 75.82 | 79.74 | 78.42 | 78.65 | 80.15 | 77.70 | 80.67 | 79.61 |
| | **OURS** | **85.10** | **92.44** | **87.24** | **84.62** | **93.31** | **86.77** | **88.89** | **97.84** | **90.37** |

## E.3. Minimal Constraints

In Table 12, we compared the effects of the minimal premise set constraint across all test sets of Multi-LogiEval and FOLIO. The results show that with the constraint, the overall accuracy is significantly improved.

*Table 12.* Performance comparison w/ and w/o minimal constraints on all benchmarks.

| MODEL | CONSTRAINT | 90% | | | 70% | | | 50% | | |
|---|---|---|---|---|---|---|---|---|---|---|
| | | ACC.↑ | AVG. NUM.↓ | FACT ACC.↑ | ACC.↑ | AVG. NUM.↓ | FACT ACC.↑ | ACC.↑ | AVG. NUM.↓ | FACT ACC.↑ |
| | | **MULTI-LOGIEVAL** | | | | | | | | |
| DEEPSEEK-V3.2 | w/o | 64.52 | 6.18 | 98.58 | 67.71 | 6.55 | 97.63 | 70.61 | 6.56 | **99.22** |
| | **w/** | **68.53** | **1.78** | **99.19** | **71.67** | **1.67** | **99.36** | **72.71** | **1.78** | 98.22 |
| GPT-5 | w/o | 60.86 | 8.98 | **98.81** | 63.91 | 8.48 | **98.08** | 71.12 | 9.08 | **98.75** |
| | **w/** | **66.95** | **1.66** | 97.60 | **68.24** | **1.35** | 96.95 | **72.43** | **1.26** | 95.61 |
| QWEN3-235B-A22B | w/o | 62.57 | 10.94 | 96.09 | 65.48 | 10.34 | **96.91** | 69.99 | 11.17 | **98.57** |
| | **w/** | **69.53** | **2.47** | **96.53** | **69.67** | **2.39** | 95.64 | **73.57** | **2.98** | 96.69 |
| | | **FOLIO** | | | | | | | | |
| DEEPSEEK-V3.2 | w/o | 85.28 | 7.01 | **96.56** | 86.14 | 7.44 | **98.68** | 85.82 | 8.42 | **97.52** |
| | **w/** | **90.81** | **1.89** | 94.79 | **90.27** | **2.08** | 89.42 | **90.46** | **1.85** | 81.61 |
| GPT-5 | w/o | 82.10 | 9.42 | **94.40** | 82.62 | 10.48 | **96.95** | 84.79 | 9.33 | 91.07 |
| | **w/** | **84.49** | **2.78** | 94.33 | **86.82** | **2.93** | 91.52 | **88.46** | **2.35** | **91.95** |
| QWEN3-235B-A22B | w/o | 76.29 | 11.24 | **97.65** | 77.65 | 10.81 | **97.09** | 81.75 | 12.80 | **99.22** |
| | **w/** | **85.10** | **1.78** | 81.01 | **84.62** | **1.88** | 82.47 | **88.89** | **1.90** | 80.41 |

## E.4. Solver Runtime

We also report the actual solver runtime grouped by the number of premises in Table 13. The timeout threshold is set to 4 seconds, and the timeout rate is 0% for every group. Across all groups, the average solver runtime remains close to 0.01 seconds, and even the 99th percentile is below 0.05 seconds. Therefore, we observe no solver failures caused by computational limits; the practical cost of OpenIKLR is dominated by LLM calls rather than symbolic solving.

*Table 13.* Solver runtime by premise number.

| PREMISE NUM | AVG. | P30 | P60 | P90 | P99 |
|---|---|---|---|---|---|
| 2 | 0.0087 | 0.0056 | 0.0102 | 0.0112 | 0.0210 |
| 3 | 0.0093 | 0.0059 | 0.0104 | 0.0118 | 0.0227 |
| 4 | 0.0118 | 0.0097 | 0.0110 | 0.0142 | 0.0289 |
| 5 | 0.0106 | 0.0067 | 0.0109 | 0.0150 | 0.0455 |
| $\geq 6$ | 0.0100 | 0.0069 | 0.0108 | 0.0153 | 0.0222 |

## E.5. Token Count

Table 14 shows the token consumption of our pipeline. Specifically, the translator uses 690 input tokens. In each iteration, the logic completion module consumes 615 input tokens, while the fact completion module requires 380 input tokens. In addition, the fact verification stage incurs 277 input tokens per iteration. Assuming a maximum number of iterations $K_{\max} = 4$, the total token consumption of our method amounts to 5778 tokens. By comparison, the SymbCoT method uses 5575 tokens in total. Our approach achieves significantly better performance. Moreover, when setting $K_{\max} = 1$, our method consumes substantially fewer tokens than SymbCoT, while still delivering superior results (73.10 vs. 38.34). This demonstrates that our framework attains higher effectiveness without incurring additional token overhead.

*Table 14.* Token counts.

| TRANS. | LOGICCOMP. | FACTCOMP | FACTVER | ALL ($K_{\max} = 1$) | ALL ($K_{\max}=3$) | ALL ($K_{\max}=4$) | SYMBCOT |
|---|---|---|---|---|---|---|---|
| 690 | 615 | 380 | 277 | 1962 | 4506 | 5778 | 5575 |

## F. Prompts

**Prompts for Symbolic Parsing**

Given a problem description and a question. The task is to parse the problem and the question into first-order logic formulas. The grammar of the first-order logic formula is defined as follows:
1) logical conjunction of expr1 and expr2: expr1 $\wedge$ expr2
2) logical disjunction of expr1 and expr2: expr1 $\vee$ expr2
3) logical exclusive disjunction of expr1 and expr2: expr1 $\oplus$ expr2
4) logical negation of expr1: $\neg$ expr1
5) expr1 implies expr2: expr1 $\rightarrow$ expr2
6) expr1 if and only if expr2: expr1 $\leftrightarrow$ expr2
7) logical universal quantification: $\forall$ x
8) logical existential quantification: $\exists$ x

OTHER RULES:
1. If a number in a formula contains a decimal point, rewrite it in a parser-safe format. For example, use 321e2 to represent 3.21e4 (no decimal points allowed).
2. Move all quantifiers to the front. Do not use nested quantifiers. Quantifiers are not allowed to appear inside subformulas of $\rightarrow, \vee, \wedge, \oplus, \leftrightarrow$. For example, expressions like $\forall$ y ( ... $\exists$x ... ) $\rightarrow$ ... are not allowed.
3. Do not use any symbols other than [$\oplus, \wedge, \vee, \rightarrow, \leftrightarrow, \forall, \exists, \neg$, (, ), =]. For example, symbols such as $\neq$ are not permitted.
4. The same symbol must not be used both as a predicate and as a constant. Letter case (uppercase vs lowercase) must not be used as a distinguishing feature; renaming is required. For example, $Dog(x)$ and $Animal(dog)$ are invalid, because Dog and dog are considered the same symbol.

5. Variable names must be single lowercase letters.
6. The Conclusion should not be included in the Premises.

———
Context:
All people who regularly drink coffee are dependent on caffeine. People either regularly drink coffee or joke about being addicted to caffeine. No one who jokes about being addicted to caffeine is unaware that caffeine is a drug. Rina is either a student and unaware that caffeine is a drug, or neither a student nor unaware that caffeine is a drug. If Rina is not a person dependent on caffeine and a student, then Rina is either a person dependent on caffeine and a student, or neither a person dependent on caffeine nor a student.
Question:
Rina is either a person who jokes about being addicted to caffeine or is unaware that caffeine is a drug.

Predicates:
Dependent(x) ::: x is a person dependent on caffeine. Drinks(x) ::: x regularly drinks coffee. Jokes(x) ::: x jokes about being addicted to caffeine. Unaware(x) ::: x is unaware that caffeine is a drug. Student(x) ::: x is a student.
Premises:
∀x (Drinks(x) → Dependent(x)) ::: All people who regularly drink coffee are dependent on caffeine. ∀x (Drinks(x) ⊕ Jokes(x)) ::: People either regularly drink coffee or joke about being addicted to caffeine. ∀x (Jokes(x) → ¬Unaware(x)) ::: No one who jokes about being addicted to caffeine is unaware that caffeine is a drug. (Student(rina) ∧ Unaware(rina)) ⊕ ¬(Student(rina) ∨ Unaware(rina)) ::: Rina is either a student and unaware that caffeine is a drug, or neither a student nor unaware that caffeine is a drug. ¬(Dependent(rina) ∧ Student(rina)) → (Dependent(rina) ∧ Student(rina)) ⊕ ¬(Dependent(rina) ∨ Student(rina)) ::: If Rina is not a person dependent on caffeine and a student, then Rina is either a person dependent on caffeine and a student, or neither a person dependent on caffeine nor a student.
Conclusion:
Jokes(rina) ⊕ Unaware(rina) ::: Rina is either a person who jokes about being addicted to caffeine or is unaware that caffeine is a drug.

———
Context:
Miroslav Venhoda was a Czech choral conductor who specialized in the performance of Renaissance and Baroque music. Any choral conductor is a musician. Some musicians love music. Miroslav Venhoda published a book in 1946 called Method of Studying Gregorian Chant.
Question:
Miroslav Venhoda loved music.

Predicates:
Czech(x) ::: x is a Czech person. ChoralConductor(x) ::: x is a choral conductor. Musician(x) ::: x is a musician. Love(x, y) ::: x loves y. Author(x, y) ::: x is the author of y. Book(x) ::: x is a book. Publish(x, y) ::: x is published in year y. Specialize(x, y) ::: x specializes in y.
Premises:
Czech(miroslav) ∧ ChoralConductor(miroslav) ∧ Specialize(miroslav, renaissance) ∧ Specialize(miroslav, baroque) ::: Miroslav Venhoda was a Czech choral conductor who specialized in the performance of Renaissance and Baroque music. ∀x (ChoralConductor(x) → Musician(x)) ::: Any choral conductor is a musician. ∃x (Musician(x) ∧ Love(x, music)) ::: Some musicians love music. Book(methodOfStudyingGregorianChant) ∧ Author(miroslav, methodOfStudyingGregorianChant) ∧ Publish(methodOfStudyingGregorianChant, year1946) ::: Miroslav Venhoda published a book in 1946 called Method of Studying Gregorian Chant.
Conclusion:
Love(miroslav, music) ::: Miroslav Venhoda loved music.
———
Context:
${context}
Question:

${question}
____

Return **only** Predicates, Premises, and Conclusion. Do not include unnecessary symbols such as "**", "numbered lists (like 1., 2., 3., 4.)", " - " etc.

---

**Prompts for Initial Premise Completion (Triggered by Logic Verification Failure)**

This task is to minimally complete the missing logic information so that the solver can successfully infer the conclusion as TRUE. The current logic formula is incomplete to support a successful proof (e.g., the solver returns UNKNOWN or FALSE).

Please generate as few additional logical information as possible (at least 1 and at most 4). Each set should contain:

- one natural language sentence, and its corresponding first-order logic formula.

These should:

1. Express real-world or commonsense knowledge;

2. Be semantically relevant to the given context and conclusion, and help connect existing predicates and entities;

3. Be useful for enabling the solver to infer TRUE and complete the proof.

Input:

1. Natural language context and conclusion;

2. Symbolic language context and conclusion;

3. The solver's proof based on the current symbolic formulas.

Context:
${context}

Conclusion:
${conclusion}

Symbolic Language Translation:
${translation}

Proof:
${reasoning}

__________

The grammar of first-order logic is defined as follows:
logical conjunction: expr1 ∧ expr2
logical disjunction: expr1 ∨ expr2
logical exclusive disjunction: expr1 ⊕ expr2
logical negation: ¬expr1
expr1 implies expr2: expr1 → expr2
expr1 if and only if expr2: expr1 ↔ expr2
logical universal quantification: ∀ x
logical existential quantification: ∃ x

__________

Output format:

logic formula ::: natural language sentence (one set per line)

__________

Example:
Context:
All squares are four-sided.

Conclusion:
All squares are shapes.

Symbolic Language Translation:
Predicates:
Square(x) ::: x is a square.
FourSided(x) ::: x is four-sided.
Shape(x) ::: x is a shape.
Premises:
$\forall x$ (Square(x) $\rightarrow$ FourSided(x))
Conclusion:
$\forall x$ (Square(x) $\rightarrow$ Shape(x))

Proof:
trying to prove original conclusion:
1 (all x (Square(x) $\rightarrow$ FourSided(x))) # label(non_clause). [assumption].
2 (all x (Square(x) $\rightarrow$ Shape(x))) # label(non_clause) # label(goal). [goal].
3 Square(x) | FourSided(x). [clausify(1)].
4 Square(c1). [deny(2)].
5 Shape(c1). [deny(2)].
6 Derived: FourSided(c1). [resolve(4,a,3,a)].
Search terminated, no contradiction found
trying to prove negation of original conclusion:
1 (all x (Square(x) $\rightarrow$ FourSided(x))) # label(non_clause). [assumption].
2 (all x (Square(x) $\rightarrow$ Shape(x))) # label(non_clause) # label(goal). [goal].
3 Square(x) | FourSided(x). [clausify(1)].
4 Square(x) | Shape(x). [deny(2)].
Search terminated, no contradiction found
So: Unknown

Example Output:
$\forall x$ (FourSided(x) $\rightarrow$ Shape(x)) ::: All four-sided things are shapes.
__________

Do not include unnecessary symbols such as "**", "numbered lists (like 1., 2., 3., 4.)", " - " etc.

---

**Prompts for Iterative Premise Completion (Triggered by Logic Verification Failure)**

This task is to minimally complete the missing logic information so that the solver can successfully infer the conclusion as TRUE. The current logic formula is incomplete to support a successful proof (e.g., the solver returns UNKNOWN or FALSE).

Please generate as few additional logical information as possible (at least 1 and at most 4). Each set should contain:

• one natural language sentence, and its corresponding first-order logic formula.

These should:

1. Express real-world or commonsense knowledge;

2. Be semantically relevant to the given context and conclusion, and help connect existing predicates and entities;

3. Be useful for enabling the solver to infer TRUE and complete the proof.

Input:

1. Natural language context and conclusion;

2. Symbolic language context and conclusion;

3. The solver's proof based on the current symbolic formulas.

Context:
${context}

Conclusion:
${conclusion}

Symbolic Language Translation:
${translation}

Proof:
${init_reasoning}

Note: The symbolic formula below is known to be incorrect.
You must output different logical information.
${failed_formula}
${failed_reasoning}

__________

The grammar of first-order logic is defined as follows:
logical conjunction: expr1 ∧ expr2
logical disjunction: expr1 ∨ expr2
logical exclusive disjunction: expr1 ⊕ expr2
logical negation: ¬ expr1
expr1 implies expr2: expr1 → expr2
expr1 if and only if expr2: expr1 ↔ expr2
logical universal quantification: ∀ x
logical existential quantification: ∃ x

__________

Output format:
logic formula ::: natural language sentence (one set per line)

__________

Example:
Context:
Some show airing at 8 pm on Monday gives out roses on TV.
If a show gives out roses on TV, then the show is an episode of The Bachelor.
All shows portraying the lives of real people are reality TV shows.
Breaking Bad is a show.
Breaking Bad is not a reality TV show.

Conclusion:
If roses are given out during Breaking Bad, then it is on Monday at 8 pm.

Symbolic Language Translation:
Predicates:
Show(x) ::: x is a show.
Airing(x, day, time) ::: x airs on day at time.
GivesOutRoses(x) ::: x gives out roses on TV.
EpisodeOf(x, y) ::: x is an episode of y.
PortrayingRealLives(x) ::: x portrays the lives of real people.
RealityTV(x) ::: x is a reality TV show.
Premises:
$\exists x$ (Show(x) $\wedge$ Airing(x, monday, eightPM) $\wedge$ GivesOutRoses(x)) ::: Some show airing at 8 pm on Monday gives out roses on TV.
$\forall x$ (GivesOutRoses(x) $\rightarrow$ EpisodeOf(x, theBachelor)) ::: If a show gives out roses on TV, then the show is an episode of The Bachelor.
$\forall x$ (PortrayingRealLives(x) $\rightarrow$ RealityTV(x)) ::: All shows portraying the lives of real people are reality TV shows.
Show(breakingBad) ::: Breaking Bad is a show.
$\neg$RealityTV(breakingBad) ::: Breaking Bad is not a reality TV show.
Conclusion:
GivesOutRoses(breakingBad) $\rightarrow$ Airing(breakingBad, monday, eightPM) ::: If roses are given out during Breaking Bad, then it is on Monday at 8 pm.

Proof:
trying to prove original conclusion:
1 (exists x (Show(x) & Airing(x,Monday,EightPM) & GivesOutRoses(x))) # label(non_clause). [assumption].
2 (all x (GivesOutRoses(x) $\rightarrow$ EpisodeOf(x,TheBachelor))) # label(non_clause). [assumption].
3 (all x (PortrayingRealLives(x) $\rightarrow$ RealityTV(x))) # label(non_clause). [assumption].
4 GivesOutRoses(BreakingBad) $\rightarrow$ Airing(BreakingBad,Monday,EightPM) # label(non_clause) # label(goal). [goal].
5 Show(c1). [clausify(1)].
6 Airing(c1,Monday,EightPM). [clausify(1)].
7 GivesOutRoses(c1). [clausify(1)].
8 GivesOutRoses(x) | EpisodeOf(x,TheBachelor). [clausify(2)].
9 PortrayingRealLives(x) | RealityTV(x). [clausify(3)].
10 Show(BreakingBad). [assumption].
11 RealityTV(BreakingBad). [assumption].
12 GivesOutRoses(BreakingBad). [deny(4)].
Search terminated, no contradiction found

Trying to prove negation of original conclusion:
1 (exists x (Show(x) & Airing(x,Monday,EightPM) & GivesOutRoses(x))) # label(non_clause). [assumption].
2 (all x (GivesOutRoses(x) $\rightarrow$ EpisodeOf(x,TheBachelor))) # label(non_clause). [assumption].
3 (all x (PortrayingRealLives(x) $\rightarrow$ RealityTV(x))) # label(non_clause). [assumption].
4 (GivesOutRoses(BreakingBad) $\rightarrow$ Airing(BreakingBad,Monday,EightPM)) # label(non_clause) # label(goal). [goal].
5 Show(c1). [clausify(1)].
6 Airing(c1,Monday,EightPM). [clausify(1)].
7 GivesOutRoses(c1). [clausify(1)].
8 GivesOutRoses(x) | EpisodeOf(x,TheBachelor). [clausify(2)].
9 PortrayingRealLives(x) | RealityTV(x). [clausify(3)].
Search terminated, no contradiction found

So: Unknown

Note: The symbolic formula below is known to be incorrect.
You must output different logical information.
∀x (EpisodeOf(x, theBachelor) → PortrayingRealLives(x)) ::: All episodes of The Bachelor portray the lives of real people.
∀x (EpisodeOf(x, theBachelor) → RealityTV(x)) ::: All episodes of The Bachelor are reality TV shows.
∀x (RealityTV(x) → ¬(x = breakingBad)) ∨ ∀x (EpisodeOf(x, theBachelor) → Airing(x, monday, eightPM)) ::: Either no reality TV show is Breaking Bad, or all episodes of The Bachelor air on Monday at 8 pm.
Proof has syntax error

Example Output:
∀x (EpisodeOf(x, theBachelor) → Airing(x, monday, eightPM)) ::: All episodes of The Bachelor air on Monday at 8 pm.
——————

Do not include unnecessary symbols such as "**", "numbered lists (like 1., 2., 3., 4.)", " - " etc.

---

**Fact Verification**

Your task is to determine whether each of the given sentences describes facts that are likely true in the real world.
A sentence is considered TRUE if:

- It describes a fact, tendency, or relationship that is generally accepted as plausible or commonly observed in the real world.

- It reflects everyday knowledge, typical human experience, or widely held beliefs, even if the statement allows for exceptions or is not precisely verifiable.

- It expresses a reasonable causal, functional, or semantic relationship that aligns with how things usually work in real-world contexts.

- For logical inference statements, the connection between antecedent and consequent should be intuitively grounded in real-world understanding, not purely arbitrary.

A sentence is considered NOT TRUE if:

- It involves clearly fictional, impossible, or nonexistent entities.

- It directly contradicts well-established real-world facts.

- It makes a strong universal claim that is clearly false in general.

- It asserts a specific factual claim that is highly implausible or unsupported by common real-world knowledge.

Input:
Premises are a list of sentences. Each sentence should be evaluated independently.

Premises:
${premise}

Output Format:
True/False (one premise per line)

Return **only** True/False.
Do not include unnecessary symbols such as "**", "numbered lists (like 1., 2., 3., 4.)", " - ", etc.

---

**Prompts for Initial Premise Completion (Triggered by Fact Verification Failure)**

You are a knowledge and logic editor for logic reasoning. Please minimally revise the non-factual premises and generate factual premises. These should:

- Real-world validity: replace all non-factual sentences (i.e., sentences that are false, unreasonable, violate common sense or natural laws, or involve fictional entities).

- Enable the proof to be TRUE.

- Be semantically relevant to the given context and conclusion, and help connect existing predicates and entities.

Input:

1. Natural language context and conclusion;

2. Symbolic language context and conclusion;

3. The solver's proof based on the current symbolic formulas;

4. Non-factual premises in natural language and symbolic language.

Context:
${context}

Conclusion:
${conclusion}

Symbolic Language Translation:
${translation}

Proof:
${reasoning}

Non-factual premises:
${sentences}

---------

The grammar of first-order logic is defined as follows:
logical conjunction: expr1 ∧ expr2
logical disjunction: expr1 ∨ expr2
logical exclusive disjunction: expr1 ⊕ expr2
logical negation: ¬expr1
expr1 implies expr2: expr1 → expr2
expr1 if and only if expr2: expr1 ↔ expr2
logical universal quantification: ∀ x
logical existential quantification: ∃ x

Output format:
logic formulas ::: natural language sentence (one set per line)

---------

Example Output:
Dependent(x) ::: x is a person dependent on caffeine
∀x (Drinks(x) → Dependent(x)) ::: All people who regularly drink coffee are dependent on caffeine.

__________
Do not include unnecessary symbols such as "**", "numbered lists (like 1., 2., 3., 4.)", " - ", etc.

---

**Prompts for Iterative Premise Completion (Triggered by Fact Verification Failure)**

You are a knowledge and logic editor for logic reasoning. Please minimally revise the non-factual premises and generate factual premises. These should:

- Real-world validity: replace all non-factual sentences (i.e., sentences that are false, unreasonable, violate common sense or natural laws, or involve fictional entities).

- Enable the proof to be TRUE.

- Be semantically relevant to the given context and conclusion, and help connect existing predicates and entities.

Input:

1. Natural language context and conclusion;

2. Symbolic language context and conclusion;

3. The solver's proof based on the current symbolic formulas;

4. Non-factual premises in natural language and symbolic language.

Context:
${context}

Conclusion:
${conclusion}

Symbolic Language Translation:
${translation}

Proof:
${reasoning}

Non-factual premises:
${sentences}

__________

The grammar of first-order logic is defined as follows:
logical conjunction: expr1 ∧ expr2
logical disjunction: expr1 ∨ expr2
logical exclusive disjunction: expr1 ⊕ expr2
logical negation: ¬ expr1
expr1 implies expr2: expr1 → expr2
expr1 if and only if expr2: expr1 ↔ expr2
logical universal quantification: ∀ x
logical existential quantification: ∃ x

Output format:
logic formulas ::: natural language sentence (one set per line)

__________

Example Output:

Dependent(x) ::: x is a person dependent on caffeine

$\forall$x (Drinks(x) $\rightarrow$ Dependent(x)) ::: All people who regularly drink coffee are dependent on caffeine.

__________

Do not include unnecessary symbols such as "**", "numbered lists (like 1., 2., 3., 4.)", " - ", etc.

