# OpenReview forum: "Open-World LLM Logical Reasoning"
_ICML.cc/2026/Conference — ICML 2026 regular_

### Official Review · Reviewer_h6c2 · 2026-02-15

**Soundness:** 4
**Presentation:** 4
**Significance:** 4
**Originality:** 4
**Overall Recommendation:** 5
**Confidence:** 3

**Summary:**

This paper targets open-world logical reasoning where the given premises are insufficient because key commonsense or real-world facts are missing. It proposes OpenIKLR, which translates natural language into first-order logic, uses a theorem prover to detect proof failure, and iteratively completes missing premises guided by solver feedback while preferring minimal additions. Added premises are accepted only after dual verification, with the solver checking sufficiency and an LLM checking factuality. Experiments on FOLIO and Multi-LogiEval under varying incomplete-premise ratios, plus a small legal case study, show consistent gains over CoT, logic baselines, and RAG variants across multiple LLM backbones.

**Compliance With Llm Reviewing Policy:**

Affirmed.

**Final Justification:**

The paper presents a solid and original neuro-symbolic approach to open-world logical reasoning, with strong empirical results across benchmarks and model backbones. My main concerns were about the realism of the open-world setup, fact verification reliability, and legal generalization, and the rebuttal addressed them well with additional experiments and analysis. I therefore maintain my Accept recommendation.

**Key Questions For Authors:**

1. How often does the NL to FOL translation fail or distort meaning?
2. In the fact verification stage, is the verifier the same backbone as the premise generator? And what is the false accept rate on deliberately wrong premises?
3. Based on my current understanding, when symbolic verification succeeds repeatedly but factual verification fails, the algorithm can return False. How often does this happen?

**Limitations:**

It would help to explicitly discuss how synthetic masking and LLM-based fact verification can bias the evaluation, and to quantify failure modes such as mistranslation, solver brittleness, and verifier false positives.

**Strengths And Weaknesses:**

I think this is a solid neuro-symbolic pipeline for incomplete-knowledge reasoning with strong empirical results. The method looks interesting and the story makes sense.

Strengths:
- Clear motivation for open-world reasoning and a clean diagnosis of why relevance-based RAG often retrieves helpful-sounding but logically useless facts.
- The solver-guided iterative completion is well structured and makes good use of proof feedback to target logically necessary missing links. Dual verification with symbolic checking plus factual checking is a sensible design that reduces purely pattern-matched premise injection.
- Experiments are fairly thorough. The authors consider multiple model families, open-world ratios, scale effects, ablations, and sensitivity to the iteration limit.

Weaknesses:
- The open-world setting is created via LLM-based premise classification and masking, so it is unclear how well it matches naturally missing knowledge in real tasks. Perhaps any downstream task benefits will be outstanding.
- Factual verification relies on an LLM without external grounding, so verifier errors and correlated hallucinations could still pass and inflate results.
- The legal experiment is promising but limited in scope and baselines, so its real-world generality is hard to judge from the current setup.

---

> ### Author Rebuttal · Authors · 2026-03-31
>
> Thank you for your thoughtful review and valuable questions! We address your questions point-to-point in the following.
>
> >**W1: Open-world setting**
> - We have a downstream legal task in the original manuscript, which shows the effectiveness of our method. We also add another dataset LegalBench (see response to W3).
> - We add an experiment using LLMs and human annotators to classify which premises are commonsense, showing **high agreements (>92%) and cross-model validity**.
> ||Claude-4.5|Doubao-2|Gemini-3|GPT-4.1|Qwen-Max|Human|
> |-|-|-|-|-|-|-|
> |Claude-4.5|-|96.21|94.35|95.68|95.94|96.78|
> |Doubao-2||-|92.66|91.744|92.88|93.66|
> |Gemini-3|||-|93.82|93.25|94.80|
> |GPT-4.1||||-|93.29|94.66|
> |Qwen-Max|||||-|93.51|
> ||
>
> - On the human-annotated dataset, we add an experiment using the 90% open-world subset, with Deepseek-V3.2 as the model.
> - The result is very similar to the original result on GPT-4.1 annotated dataset.
> - Still on the human verified dataset, our method stably outperforms the baselines.
> |||Acc|Recall|F1|
> |-|-|-|-|-|
> |GPT-4.1|CoT|83.36|76.99|83.52|
> ||Ours|90.81|94.12|91.82|
> |Human|CoT|83.47|76.20|82.91|
> ||**Ours**|**91.37**|**94.79**|**91.15**|
> ||
>
> >**W2: Factual verification relies on an LLM**
>
> **Response:** Thanks for the helpful suggestion. We agree that fact verification may incur errors because relying on an LLM without external grounding. In such cases, **an error in factual judgment often leads to an incorrect final output** (since only when both logical verification and factual verification are correct, the final result could match the ground truth label). Therefore, **the potential unreliability of factual verification may actually reduce the overall accuracy of the final output.**
>
> >**W3: The legal experiment is promising but limited in scope and baselines**
>
> **Response:** Thanks for the feedback. We **added experiments on LegalBench [1], which is an English-language legal reasoning benchmark based on U.S. law**. We selected the SARA Entailment dataset, retaining the “description” and “question” for the QA task. The results are as follows.
>
> |Model|Method|Acc|Recall|F1|
> |-|-|-|-|-|
> |DeepSeek-V3.2|Direct|52.94|28.68|37.86|
> ||CoT|56.99|43.38|50.21|
> ||**Ours**|**59.36**|**60.23**|**58.17**|
> |GPT-5|Direct|55.15|**86.76**|65.92|
> ||CoT|63.60|82.21|73.92|
> ||**Ours**|**66.72**|83.58|**76.37**|
> |Claude-Sonnet-4.5|Direct|53.31|16.18|25.73|
> ||CoT|64.34|51.47|59.07|
> ||**Ours**|**68.93**|**66.24**|**67.81**|
> |Qwen3-235B-A22B|Direct|52.21|**64.71**|57.52|
> ||CoT|58.67|39.71|49.09|
> ||**Ours**|**61.75**|54.69|**58.21**|
> ||
>
> Based on the above results, we found that our method outperforms the baselines **across real-world legal application scenarios involving different countries, languages, and tasks**.
>
> [1] Guha, N., et al. (2023). LegalBench: A Collaboratively Built Benchmark for Measuring Legal Reasoning in Large Language Models. In NeurIPS.
>
> >**Q1: NL to FOL translation quality**
>
> **Response:** Thanks for the question. We evaluate on FOLIO, one of the few datasets with human-annotated FOL formulas, i.e., the gold/standard translation results. For each example we compare our FOL translation with the gold FOL to examine the translation accuracy.
>
> |Model|FOLIO|
> |-|-|
> |Qwen3-235B-A22B|86.86|
> |GPT-5|86.47|
> |DeepSeek-V3.2|88.62|
> ||
>
> >**Q2: In the fact verification stage, is the verifier the same backbone as the premise generator? And what is the false accept rate on deliberately wrong premises?**
>
> **Response:** Thanks for the suggestion. Yes, it is the same backbone. We randomly selected 300 cases, manually labeled the truthfulness of each generated premise, and then analyzed the fact verifier 's (the LLM) judgments.
>
> ||Acc|Prec|Recall|F1|
> |-|-|-|-|-|
> |Fact Verifier|94.75|95.83|98.53|95.71|
> ||
>
> >**Q3: How often about the situation that symbolic verification succeeds repeatedly but factual verification fails.**
>
> **Response:** Thanks for the question. We consider the  iteration count $k_{max}=4$ as an example, and provide the following analysis of this issue. When $k=1$, a rate of 79.95% indicates that: after the first round of completion, samples that **passed the logical verification but failed the factual verification** account for 79.95% of the samples that **passed the logical verification in this round**.
>
> | Iteration Count $k$|Rate|
> |-|-|
> |1|79.95%|
> |2|68.07%|
> |3|51.76%|
> |4|44.11%|
> ||
>
> ---
>
> Please let us know if we have properly addressed your questions and we are more than happy to discuss more!

---

> > ### Author Rebuttal · Reviewer_h6c2 · 2026-04-01
> >
> > Thanks to the authors for their response. I believe my questions have been largely addressed, as the authors provided very detailed experiments across a wide range of backbone models. I also believe that some limitations do exist; however, they seem more like side effects of the method, which are largely inevitable and, in some sense, accompany the strengths presented in the paper. Considering the results and the other responses as well, the paper generally makes sense to me, and I find its contributions noteworthy. Therefore, I would recommend accepting this paper at the conference. I will keep my score unchanged, since I am already recommending it as a technically solid paper worthy of acceptance.

---

> > > ### Author Response · Authors · 2026-04-01
> > >
> > > Dear Reviewer h6c2,
> > >
> > > We really appreciate that your concerns have been adequately addressed. Thanks for your recognition of our **_very detailed experiments across a wide range of backbone models_**. Moreover, we feel encouraged that you consider our **_contributions noteworthy_**, and recommending our paper as **_a technically solid paper worthy of acceptance_**.
> > >
> > > Many thanks,
> > >
> > > Authors of #22398

---

### Official Review · Reviewer_6Xs2 · 2026-03-10

**Soundness:** 3
**Presentation:** 3
**Significance:** 3
**Originality:** 3
**Overall Recommendation:** 3
**Confidence:** 4

**Summary:**

This paper proposes OpenIKLR, an Open-world Incomplete-Knowledge-aware Logical Reasoning framework that addresses the "reasoning gap" when provided premises are insufficient to derive a conclusion. The work appears to analyze a broad context of logical reasoning under incomplete information, moving beyond the closed-world assumption. The framework translates natural language into first-order logic via an LLM, uses a symbolic solver (Prover9) to identify reasoning gaps, and iteratively generates a minimal set of missing premises through LLM completion. Experiments on Multi-LogiEval, FOLIO, and a Chinese legal dataset (CAIL2024) across five LLM backbones show consistent improvements over baselines.

**Compliance With Llm Reviewing Policy:**

Affirmed.

**Key Questions For Authors:**

Q1: How do you ensure the correctness of ground-truth labels after masking commonsense premises?
Consider a concrete example: suppose the original closed-world problem has premises {P1: "All metals conduct electricity", P2: "Things that do not conduct electricity are insulators", P3: "Iron is a metal"}, with the conclusion "Iron is an insulator" labeled as False. If P1 is identified as commonsense and masked, the remaining premises (P2, P3) are logically insufficient to derive either True or False for the conclusion — the strictly correct answer should be Unknown, not False. Yet the dataset still retains the original label False, meaning a system that honestly answers "Unknown" would be penalized, while only a system that successfully recovers the masked premise would be rewarded. This raises two concerns: (1) How many samples in your benchmark suffer from such label shifts after masking, where the logically correct answer under the open-world setting differs from the retained closed-world label? (2) Since the commonsense identification is performed by GPT-4.1 without human validation, how do you ensure that all masked premises are indeed recoverable commonsense rather than domain-specific knowledge that a reasoning system cannot be expected to infer? Without analyzing the label consistency before and after masking, the reported accuracy improvements may be inflated by label noise in the constructed benchmark.

Q2: What is the actual wall-clock time per sample, and how often does Prover9 timeout? How sensitive is performance to the solver's timeout threshold?

**Limitations:**

The framework inherits the well-known fragility of NL-to-FOL translation — parsing errors in the symbolic representation propagate through the entire pipeline and cannot be self-corrected by the solver. The reliance on Prover9 limits the framework to FOL, excluding probabilistic, temporal, or defeasible reasoning common in real-world scenarios. The approach assumes the LLM has sufficient world knowledge to generate correct missing premises, which may fail in highly specialized or low-resource domains where the LLM's parametric knowledge is sparse.

**Strengths And Weaknesses:**

Strengths:
1. Well-motivated problem formulation. The distinction between closed-world and open-world reasoning is practically important and clearly articulated.
2. Comprehensive experiments. The paper evaluates across 5 LLM backbones, 3 datasets, 3 open-world proportions, 7+ baselines spanning different paradigms (CoT, symbolic, RAG), and multiple model scales (8B to 235B).
3. Practical applicability. The legal domain experiment (CAIL2024) demonstrates real-world utility beyond synthetic benchmarks.

Weaknesses:
1. Handling of "False" and "Unknown" labels is asymmetric. The algorithm (Algorithm 1) is biased toward predicting "True" — it returns True as soon as one iteration passes both verifiers, but only returns False after exhausting all iterations and checking n>Kmax/2. For samples where the ground truth is False or Unknown, the framework may over-generate premises to force a proof, leading to inflated recall at the cost of precision. There is a need to examine the distribution of Precision and Recall for each categrory (True, False, and Unknown).
2. Scalability of the symbolic solver. Prover9 is used for FOL theorem proving, which has worst-case exponential complexity. The paper acknowledges its complexity but does not report actual solver runtimes, timeout rates, or how often the solver fails due to computational limits rather than logical insufficiency.
3. Limited analysis of failure modes. There is no qualitative error analysis showing when and why OpenIKLR fails — e.g., cases where the LLM generates logically valid but factually wrong premises that pass verification, or where symbolic parsing errors cascade into incorrect completions.

---

> ### Author Rebuttal · Authors · 2026-03-31
>
> Thank you for your valuable comments! We will response point to point.
> > W1: Asymmetric False and Unknown
> - Binary Real-World Setting (No "Unknown"): Our open-world setting is a strict **binary classification** task. In practical, high-stakes domains like legal judgment, a system must reach a definitive answer from True and False.
> - Fact Verifier Prevents "Forced Proofs": Our method does not inflate Recall by over-generating fake premises to "force" a True prediction. Any premise generated to bridge the reasoning gap **must pass the Fact Verifier**. If the ground truth is False, the logically required premises will violate real-world facts, get rejected by the LLM, and **correctly cause the iteration to exhaust and return False**.
> - To further address the concerns, we present the class-wise breakdown on FOLIO using DeepSeek-V3.2.
> ||Acc|Prec|Recall|F1|
> |-|-|-|-|-|
> |True|90.32|89.20|93.83|91.46|
> |False|90.32|91.88|86.00|88.84|
> ||
>
> > W2: Scalability of the symbolic solver & Q2: Timeout rate
> - For different reasoning depths and numbers of premises, we **add experiments with DeepSeek-V3.2 on Multi-LogiEval**, reporting the actual solver runtimes and timeout rates. The timeout threshold is set to 4 second.
> |Reasoning Depth|Avg|P60|P90|P99|
> |-|-|-|-|-|
> |1|0.0091|0.0103|0.0113|0.0171|
> |2|0.0086|0.0101|0.0111|0.0222|
> |3|0.0097|0.0107|0.0121|0.0182|
> |4|0.0104|0.0109|0.0132|0.0401|
> |5|0.0118|0.0111|0.0158|0.0289|
> ||
>
> |Premise Number|Avg|P60|P90|P99|
> |-|-|-|-|-|
> |2|0.0087|0.0102|0.0112|0.0210|
> |3|0.0093|0.0104|0.0118|0.0227|
> |4|0.0118|0.0110|0.0142|0.0289|
> |5|0.0106|0.0109|0.0150|0.0455|
> |≥6|0.0100|0.0108|0.0153|0.0222|
> ||
> - We also **report performance under different Timeout threshold** and find that **no solver failures from computational limits**.
>
> |Timeout Threshold|Acc|Recall|F1|Timeout Rate|
> |-|-|-|-|-|
> |1|68.10|68.70|77.99|0|
> |2|68.53|68.70|78.22|0|
> |4|68.81|69.57|78.59|0|
> |6|67.53|68.00|77.50|0|
> |10|68.38|69.39|78.85|0|
> ||
>
> > W3: Failure modes
>
> We have manually inspected the failure cases from our experiments and **categorized them into four failure modes**.
> - NL2SL\_Err: Initial translation errors between NL and SL
> - Comp\_Err: Premise completion errors
> - SL2NL\_Err: Errors occur when translating completed SL back to NL
> - FactVer\_Err: Fact Verification errors
> |NL2SL\_Err|Comp\_Err|SL2NL\_Err|FactVer\_Err|
> |-|-|-|-|
> |78.54%|4.08%|13.38%|4.00%|
>
> > Q1: Correctness of ground-truth labels after masking commonsense
>
> Thanks for the comment. We were wondering if there was a misreading–the exact scenario you described is the fundamental motivation for our framework and illustrates the precise **difference between closed-world and open-world reasoning**.
> 1. Label Shifts
> - In an open-world setting, the ground-truth label reflects the objective reality based on the explicitly provided premises **combined with implicit real-world commonsense**. The label **does not "shift" to Unknown** just because a commonsense premise is omitted from the text.
> - **Using your excellent example**: A traditional closed-world system, looking only at P2 and P3, will indeed output "Unknown" because it lacks the capacity to bridge the gap. This is not a feature; it is a failure mode in open-world applications. Our OpenIKLR framework is designed exactly to overcome this by recognizing that P1 is missing, generating it, and correctly concluding "False."
> - Therefore, a system that answers "Unknown" is not being unfairly penalized for being honest; it is simply failing the open-world reasoning task by behaving like a fragile closed-world solver. Because the ground truth remains anchored to real-world validity, there are **no label shifts** in our benchmark.
> 2. Valid Commonsense
>
> We **add experiments using LLMs and human annotators** to classify which premises are commonsense, showing high agreements (>92%) and cross-model validity.
> ||Claude-4.5|Doubao-2|Gemini-3|GPT-4.1|Qwen-Max|Human|
> |-|-|-|-|-|-|-|
> |Claude-4.5|-|96.21|94.35|95.68|95.94|96.78|
> |Doubao-2||-|92.66|91.74|92.88|93.66|
> |Gemini-3|||-|93.82|93.25|94.80|
> |GPT-4.1||||-|93.29|94.66|
> |Qwen-Max|||||-|93.51|
> ||
> - On the human-annotated dataset, **we add an experiment using the 90% open-world subset**, with Deepseek-V3.2 as the model.
> - The result is very similar to the original result on GPT-4.1 annotated dataset.
>
> |||Acc|Recall|F1|
> |-|-|-|-|-|
> |GPT-4.1|CoT|83.36|76.99|83.52|
> ||Ours|90.81|94.12|91.82|
> |Human|CoT|83.47|76.20|82.91|
> ||Ours|91.37|94.79|91.15|
> ||
>
> > Limitation: NL-to-FOL translatio
>
> We **add experiments to evaluate the accuracy of FOL formalization** on FOLIO. For each sample we compare our FOL translation with the gold FOL to examine the translation accuracy.
>
> |Model|FOLIO|
> |-|-|
> |Qwen3-235B-A22B|86.86|
> |GPT-5|86.47|
> |DeepSeek-V3.2|88.62|
> ||
>
> From above, we find that **natural language can be correctly translated into FOL with high accuracy**.
> ***
> We are eager to hear your feedback. We’d deeply appreciate it if you could let us know whether your concerns have been addressed.

---

### Official Review · Reviewer_wXS1 · 2026-03-11

**Soundness:** 2
**Presentation:** 3
**Significance:** 3
**Originality:** 3
**Overall Recommendation:** 3
**Confidence:** 3

**Summary:**

This paper studies the problem of logical reasoning under incomplete knowledge in open-world settings. The authors point out that existing logical reasoning methods default to the closed-world assumption, and propose OpenIKLR to address this: first translating natural language into symbolic representations, then using an external solver to check for reasoning gaps; if gaps exist, an LLM iteratively completes the minimal necessary premises; once the solver verifies the logic, the added premises are passed to a fact verifier for factual verification. The method claims to outperform direct answering, CoT, RAG, and several logical reasoning baselines on multiple logical reasoning benchmarks and one legal dataset.

**Compliance With Llm Reviewing Policy:**

Affirmed.

**Key Questions For Authors:**

1. Theorem 3.2 proves that the PAC-Bayes upper bound becomes looser as the number of premises increases. How do the authors translate this into theoretical support for the minimal premise set? Where is the relationship between a looser bound and worse true risk demonstrated? What is the concrete construction of the posterior Q?

2. The open-world test set is constructed by GPT-4.1 classifying and masking "commonsense premises," and the method itself completes exactly this type of premise. How do the authors rule out the circularity bias between the test set construction and the method's assumptions? Is there cross-model consistency verification or human spot-checking?

3. Why is fact verification not executed immediately after each newly generated premise? Has there been any experimental comparison between incremental verification and the current post-hoc design in terms of efficiency? Under what conditions does the current design become significantly inefficient?

4. Why does the RAG baseline not include stronger retrieval configurations? How sensitive are the current negative conclusions about RAG to this particular configuration choice?

**Limitations:**

No. The most critical limitation — the framework's fundamental dependence on accurate FOL formalization — is mentioned only in passing in the Impact Statement and is absent from any dedicated discussion in the main text. The authors should explicitly discuss the failure modes introduced by FOL translation errors, the method's restricted applicability to highly structured domains, and scenarios where missing knowledge cannot be reduced to a crisp set of FOL premises.

**Strengths And Weaknesses:**

Strengths:

The problem definition is clear, and the distinction between closed-world and open-world reasoning is practically meaningful. The framework is highly modular, compatible with closed-source models as an inference-time plug-in without modifying model parameters. The experiments cover a wide range of backbone models, and the ablation results generally support the contribution of each module. The legal domain application is a convincing case, though it may suggest that the method's applicability is largely confined to highly structured domains.

Weaknesses:

W1. The theoretical analysis suffers from a motivational misalignment; the connection to the method design requires clearer justification.

The PAC-Bayes bound in Section 3.5 (Theorem 3.1) is technically unproblematic, but the interpretation of Theorem 3.2 is questionable. What the theorem proves is: under a given cardinality-aware prior, the PAC-Bayes upper bound becomes looser as the expected number of completed premises increases — this is a monotonicity result about the bound. The authors use this as a "principled justification" for the minimal premise set, but there is a gap here: a looser upper bound does not directly imply worse true risk, and the authors do not discuss the relationship between the two. Furthermore, the paper does not explain how the posterior Q is defined from the LLM's iterative completion process, only stating vaguely that the posterior can "arbitrarily depend on the sample." Taken together, this theoretical analysis looks more like post-hoc motivation for a design choice (the minimal premise constraint) rather than a theoretical foundation that actually guides the method. At minimum, the authors should address this gap.

W2. The construction of the open-world test sets is deeply coupled with the method’s own assumptions, and the paper does not adequately discuss this.

The paper's open-world setting does not come from naturally incomplete scenarios — it is constructed by using GPT-4.1 to classify which premises belong to "real-world knowledge or commonsense" and masking them. The problem is: the Completion module of OpenIKLR generates exactly this type of premise, and the fact verifier also validates the factual accuracy of such knowledge, using criteria closely aligned with the GPT-4.1 classification logic used in data construction. This deep coupling between the construction logic and the method's assumptions introduces a clear circularity risk — the test set defines "missing" premises according to exactly the type of knowledge the method is best at recovering. The external validity of the conclusion "this method is effective" under such a setup is questionable. The paper provides no human verification, cross-model consistency analysis, or any discussion addressing this risk, which is a significant omission.

W3. The post-hoc placement of fact verification is a genuine efficiency concern, but the paper does not specifically analyze the cost of this design choice. The paper explicitly describes that the system only enters the fact verification stage after the solver has already found a logically valid proof path; if verification fails, that iteration is discarded and the loop continues. For candidate premises that are logically connectable but factually incorrect, the system wastes a complete iteration. To clarify: the paper is not entirely without cost analysis — Table 9 gives per-module token counts, and the Kmax sensitivity experiment provides some reference. However, these analyses characterize the overall pipeline cost, not the efficiency cost of the specific design choice of post-hoc verification versus incremental verification. Has the author tried incremental fact verification? Under what conditions does the current design become significantly inefficient? These questions are nowhere discussed in the paper.

W4. The comparison with RAG is insufficiently thorough, and the scope of the current conclusions is overstated.

The RAG baselines use a fixed configuration: Qwen3-8B-Embedding, ATOMIC2020, and Top-k=3. The paper acknowledges in related work that "RAG performance heavily depends on retrieval quality," yet no sensitivity analysis over retrieval configurations is conducted in the experimental design. Whether a larger Top-k, re-ranking, or a stronger embedding model would substantially improve RAG results has not been ruled out. The authors directly assert a performance hierarchy of RAG, CoT, RAG+OpenIKLR, OpenIKLR and claim that RAG often recalls insufficient or unnecessary premises — under the current experimental evidence, this statement is an overgeneralization.

W5. The strong dependence on FOL parseability is a fundamental limitation of the method, yet the paper does not systematically address this.

The effectiveness of the entire framework critically depends on the assumption that natural language can be correctly translated into FOL. The paper itself explicitly states in the methodology section that any error in the initial SL representation would lead to incorrect gap identification in subsequent steps. The legal domain and highly structured datasets like FOLIO and Multi-LogiEval happen to be the scenarios most amenable to FOL formalization, but in broader open-world settings, much of the missing commonsense knowledge is probabilistic and vague — forcing it into symbolic form introduces translation errors that cascade through every subsequent step. The Impact Statement discusses risks around safety-critical domains and the possibility that "incorrect or biased background knowledge may still be generated or verified imperfectly," but does not address FOL translation failure or its downstream cascading effects. This gap sits in clear tension with the paper's claimed "open-world" positioning.

---

> ### Author Rebuttal · Authors · 2026-03-31
>
> Thank you for your thoughtful review and valuable questions!
> > W1+Q1: Justification of Theorem 3.2
>
> Theorem 3.2 proves the **monotonicity of the bound** rather than the true risk. We will clarify its connection to true risk and explicitly formalize $Q$ in the revised version.
> 1. Definition of $Q$
> - The **autoregressive generation distribution** of LLM naturally defines a probability on $\mathcal{P}$.
> - In our algorithm, $Q$ is defined by conditioning the above probability on the event that the **logical chain is validated** by solver.
> 2. Looser bound to true risk
> - The monotonicity in Theorem 3.2 is for those pass the solver validation.
> - The logic solver drastically reduces the empirical risk by eliminating logically flawed reasoning paths (driving $\hat{R}\_S \to 0$).
> - In this case, a looser bound of complexity penalty term directly implies a looser bound for true risk.
> - A looser bound does not necessarily imply a worse true risk, yet this holds true in most cases.
>
> > W2+Q2: Construction of the open-world test sets
> - We **add experiments using LLMs and human annotators** to classify which premises are commonsense, showing **high agreements (>92%) and cross-model validity**.
> ||Claude-4.5|Doubao-2|Gemini-3|GPT-4.1|Qwen-Max|Human|
> |-|-|-|-|-|-|-|
> |Claude-4.5|-|96.21|94.35|95.68|95.94|96.78|
> |Doubao-2||-|92.66|91.744|92.88|93.66|
> |Gemini-3|||-|93.82|93.25|94.80|
> |GPT-4.1||||-|93.29|94.66|
> |Qwen-Max|||||-|93.51|
> - On the human-annotated dataset, we **add experiments using the 90% open-world subset**, with Deepseek-V3.2 as the model.
> - The result is very similar to the original result on GPT-4.1 annotated dataset.
> - Still on the human verified dataset, our method stably outperforms the baselines.
> |||Acc|Recall|F1|
> |-|-|-|-|-|
> |GPT-4.1|CoT|83.36|76.99|83.52|
> ||Ours|**90.81**|**94.12**|**91.82**|
> |Human|CoT|83.47|76.20|82.91|
> ||Ours|**91.37**|**94.79**|**91.15**|
> ||
>
> > W3+Q3: Post-hoc placement of fact verification
> - We **add experiments on FOLIO comparing the post-hoc fact verification with the incremental verification**.
> - We provide the performance as well as the averaged and percentiled token cost and number of API calls.
> - Results show that our post-hoc fact verification is superior to the incremental verification on **both performance and efficiency**.
> ||Avg Token|Avg API Calls|Acc|Recall|F1|
> |-|-|-|-|-|-|
> |post-hoc|3106.92|4.77|90.81|94.12|91.82|
> |incremental|3754.51|7.35|80.14|75.00|80.11
>
> > W4+Q4: Thorough comparison with RAG
> - We **add three experiments on the configuration of Top-k, embedding model and re-ranking** in the best RAG baseline SireRAG.
> |Top-K||Acc|Recall|F1|
> |-|-|-|-|-|
> |3|RAG|82.26|71.68|81.54|
> ||RAG+Ours|**86.59**|**85.84**|**87.52**|
> |5|RAG|83.20|72.86|82.61|
> ||RAG+Ours|**86.95**|**84.37**|**87.80**|
> |10|RAG|81.88|70.86|80.44|
> ||RAG+Ours|**85.24**|**83.37**|**85.86**|
> |15|RAG|81.84|69.94|80.33|
> ||RAG+Ours|**85.24**|**83.02**|**85.19**|
> |20|RAG|81.04|68.39|80.17|
> ||RAG+Ours|**85.08**|**83.02**|**85.06**|
> |N/A|Ours|**90.81**|**94.12**|**91.82**|
> ||
>
> |Embed Model||Acc|Recall|F1|
> |-|-|-|-|-|
> |Qwen3-8B-Embedding|RAG|82.26|71.68|81.54|
> ||RAG+Ours|**86.59**|**85.84**|**87.52**|
> |llama-embed-nemotron-8b|RAG|80.91|69.39|80.21|
> ||RAG+Ours|**86.91**|**87.32**|**87.96**|
> |BGE-M3|RAG|80.26|66.73|79.07|
> ||RAG+Ours|**84.49**|**81.42**|**85.19**|
> |NV-Embed-v2|RAG|79.45|64.37|77.81|
> ||RAG+Ours|**86.27**|**83.78**|**86.98**|
> |N/A|Ours|**90.81**|**94.12**|**91.82**|
> ||
>
> |Reranker||Acc|Recall|F1|
> |-|-|-|-|-|
> |None|RAG|82.26|71.68|81.54|
> ||RAG+Ours|**86.59**|**85.84**|**87.52**|
> |Qwen3-Reranker-8B|RAG|83.49|75.29|83.52
> ||RAG+Ours|**86.91**|**86.43**|**87.86**|
> |BGE-Reranker-large|RAG|85.11|77.35|85.22
> ||RAG+Ours|**87.10**|**85.88**|**87.95**|
> |N/A|Ours|**90.81**|**94.12**|**91.82**|
> ||
> - We choose the best configuration (Top-N=20, Top-k=3, Qwen3-8B-Embedding, Qwen3-Reranker-8B), and use GPT-4.1 (on the whole dataset) and human annotator (on 100 random samples) to **evaluate the sufficiency or necessity rate of the retrievals**.
> - Results show RAG is very likely to retrieve insufficient or unnecessary statements.
> ||necessity|sufficiency|GPT-4.1|Human|
> |-|-|-|-|-|
> ||√|√|4.37%|2.00%|
> ||√|×|17.65%|24.00%|
> ||×|√|9.42%|11.00%|
> ||×|×|68.56%|63.00%|
> > W5+Limitation: Dependence on FOL
>
> > Limitation: NL-to-FOL translatio
>
> Thanks for reminding us the dependency on FOL. We **add experiments to evaluate the accuracy of FOL formalization** on FOLIO, one of the few datasets with human-annotated FOL formulas, i.e., the gold/standard translation results. For each example we compare our FOL translation with the gold FOL to examine the translation accuracy.
>
> |Model|FOLIO|
> |-|-|
> |Qwen3-235B-A22B|86.86|
> |GPT-5|86.47|
> |DeepSeek-V3.2|88.62|
> ||
>
> From above, we find that **natural language can be correctly translated into FOL with high accuracy**.
>
> ***
>
> We are eager to hear your feedback. We’d deeply appreciate it if you could let us know whether your concerns have been addressed.

---

> > ### Author Rebuttal · Reviewer_wXS1 · 2026-04-04
> >
> > Thanks for your response. Although the authors have clarified some questions, the concerns are not fully addressed. I will keep my scores.

---

> > > ### Author Response · Authors · 2026-04-04
> > >
> > > Thanks for your rebuttal acknowledgement. We noted you choose ```(b) Partially resolved - I have follow-up questions for the authors``` - **could you kindly let us know what are your follow-up questions?** We also noted that you wrote, ```Although the authors have clarified some questions, the concerns are not fully addressed. I will keep my scores``` - **could you also kindly let us know what are your unaddressed concerns?**
> > >
> > > ***
> > >
> > > **In addition, we identified several misreadings in the original review and included additional analyses and experiments as follows.**
> > >
> > > >W1+Q1: Theorem 3.2
> > > - Sec. 3.5 does not aim to assert “larger completion implies larger true risk”; it offers a **regularization rationale**. Under $P(\Delta)\propto \exp(-\alpha|\Delta|)$, larger $\mathbb{E}[|\Delta|]$ increases the complexity term in the PAC‑Bayes upper bound, while the true risk itself is not directly claimed to worsen.
> > > - In our setup, the posterior $Q(\Delta)$ denotes the LLM’s completion distribution conditioned on solver acceptance, and the prior $P(\Delta)$ encodes a cardinality preference that favors shorter $\Delta$.
> > > - This implies we should prefer **minimal solver‑sufficient completions** to retain tighter generalization control and avoid unnecessary complexity inflation.
> > > - Empirically, removing minimality increases added premises **1.89→7.01** and reduces Acc/F1 **90.81/91.82→85.28/86.15** (FOLIO), matching the bound’s directionality.
> > > - For completeness, note that the bound’s monotonicity follows via the KL term $\mathrm{KL}(Q\|P)$ under a cardinality‑aware prior, which grows with expected $|\Delta|$; our solver verification reduces empirical risk, and the prior regularizes completion complexity.
> > >
> > > >W2+Q2: Open-world setting
> > > - Our setting targets the **open‑world truth value** where commonsense can be implicit; therefore masking a commonsense premise does **not** change the ground‑truth label by design.
> > > - To mitigate circularity to any single annotator model, we **cross‑validated commonsense identification with 5 LLMs and human annotators**, achieving **>92% agreement**.
> > > - We have verified that the performance trend is stable under human reconstruction, and audited factual verification on 300 sampled premises (manual truth labels).
> > > - Overall, the benchmark targets **recoverable commonsense completion**, not arbitrary naturally missing knowledge.
> > >
> > > >W3+Q3: Algorithm and efficiency
> > > - On FOLIO, we have verified in our rebuttal that **"post‑hoc verification" outperforms "incremental"**.
> > > - Observed costs of our method is less than logic‑pass fact‑fail.
> > > - In brief, our algorithm translates NL→FOL and invokes the solver; failing proofs trigger up to $K_{\max}$ **minimal** completions guided by the failed trace; successful logic then requires fact verification.
> > > - We **add more detailed efficiency analysis**:
> > >
> > > ||Init Translation|Logic Verification|Fact Verification|Label|Avg tokens|Avg API Calls|
> > > |-|-|-|-|-|-|-|
> > > |1|×|N/A|N/A|All|811.26|2|
> > > |2|√|×|N/A|True|3401.82|5|
> > > |3|√|×|N/A|False|3573.68|5|
> > > |4|√|√|×|True|2455.07|6.23|
> > > |5|√|√|×|False|2909.23|5.76|
> > > |6|√|√|√|True|1358.50|2.49|
> > > |7|√|√|√|False|2131.48|4.16|
> > > - The relative inefficiency arises in branches where logic passes but fact verification fails, particularly on **False** cases. This phenomenon is aligned with the expected behavior.
> > > >W4+Q4: comparison with RAG
> > > - Scope claim: **Retrieval quality alone does not remove the reasoning gap**.
> > > - We have verified that **standalone our method with no retrieval outperforms RAG methods.**
> > > - We **add experiments on different knowledge bases:**
> > > |Knowledge Base||Acc|Recall|F1|
> > > |-|-|-|-|-|
> > > |ATOMIC2020|RAG|82.26|71.68|81.54|
> > > ||RAG+Ours|**86.59**|**85.84**|**87.52**|
> > > |Wikipedia|RAG|81.74|69.62|80.68|
> > > ||RAG+Ours|**87.24**|**88.20**|**88.33**|
> > > |ConceptNet|RAG|78.68|63.72|76.60|
> > > ||RAG+Ours|**84.65**|**79.06**|**84.94**|
> > > |N/A|Ours|**90.81**|**94.12**|**91.82**|
> > > - The results show that **our method remains superior against RAG methods across knowledge bases.**
> > > - Besides, we **add experiments to compare more RAG methods:**
> > > |Method|Acc|Recall|F1|
> > > |-|-|-|-|
> > > |HyDE|78.68|70.50|78.36|
> > > |SireRAG|82.26|71.68|81.54|
> > > |Self-RAG|66.24|64.01|67.50|
> > > |CRAG|73.82|72.17|74.55|
> > > |Speculative RAG|78.69|75.40|77.93|
> > > |Ours|**90.81**|**94.12**|**91.82**|
> > > ||
> > > - **Our method outperforms the RAG methods, including the best standalone RAG.**
> > > >FOL dependence
> > > - We have quantified NL→FOL parseability on FOLIO, stably achieving high accuracy (> 86%) on Qwen3-235B, GPT-5, and DeepSeek-V3.2.
> > > - Our error analysis shows that **failures are dominated by NL2SL (78.54%) and SL2NL (13.38%)**, with completion (4.08%) and fact verification (4.00%) contributing less.
> > > - Therefore, OpenIKLR is best suited to domains where **symbolic formalization is sufficiently reliable** (e.g., legal), rather than arbitrary probabilistic or vague open‑world reasoning.
> > > ***
> > > We sincerely hope these additional clarifications and experiments can fully address your concerns!

---

### Official Review · Reviewer_Pg5E · 2026-03-11

**Soundness:** 3
**Presentation:** 3
**Significance:** 3
**Originality:** 3
**Overall Recommendation:** 5
**Confidence:** 3

**Summary:**

This paper claims that eixsting research in logical reasoning confined within a so-called "closed-world assumption", where it is assumed that all the necessary premises are explicitly provided for the reasoning system according to which a conclusion could be made. The paper considers that in reality, however, necessary premises are sometimes missing, or are considered as implicit commonsense knowledge that are not recorded explicitly, making it impossible to draw conclusions with broken logical reasoning chains. To enable the LLM-based reasoning systems to be able to conduct logical reason under such open-world setting, the paper proposes an iterative framework based on LLM-based fact verification and logic verifier, where the logic verifier identifies the logical gap, and the LLM provides additional facts/premises to make the logical reasoning complete. The paper justifies their proposed framework from empirical experiments that their proposed method outperforms other baselines in the open-world setting.

**Compliance With Llm Reviewing Policy:**

Affirmed.

**Key Questions For Authors:**

1. Why it has to be a minimal set? Can I just provide a lot of factual knowledge that contain all I need, and then make inference? Would that reduce time costs?
2. For more complicated logical reasoning cases, is it possible that the system iteratively updates the premises up to $K_{max}$ but still fail in logic solver? How the proposed framework deals with it?

**Limitations:**

The paper does not include a discussion of its limitations or the potential negative societal impacts of the proposed work.

**Strengths And Weaknesses:**

Strength:
1. The proposed framework in the open-world setting logical reasoning is novel;
2. The paper is clearly written and easy to understand;
3. The paper conducts a comprehensive empirical experiments to jusitify the effectiveness of the proposed method.

Weakness: It is unclear how the theoretical analysis may be connected to the pratical usage of the proposed framework. Furthermore, what assumptions are considered in the theory.

---

> ### Author Rebuttal · Authors · 2026-03-31
>
> Thank you for your thoughtful review and valuable questions! We address your questions point-to-point in the following.
>
> >**Weakness: Theoretical analysis**
> 1. Connection of the theoretical analysis to practical usage
> - According to Theorem 3.2, increasing the number of completed premises leads to a **looser PAC-Bayes upper bound**.
> - Since our logic solver effectively eliminates logically flawed reasoning paths—driving the empirical risk ($\hat{R}_S$) toward zero —**controlling the complexity penalty term $\sqrt{
> \frac{
> \mathrm{KL}(Q(\Delta)\|P(\Delta))+\log\!\frac{1}{\delta}
> }{
> 2(n-1)
> }}$ is essential**.
> - A looser bound directly implies a looser bound for true risk. While a looser bound does not unconditionally guarantee a worse true risk, it holds true in most cases. Therefore, **practically pursuing a minimal set controls this risk** of logical derailment.
>
> 2. Assumption considered
>
> In our revised manuscript, we will explicitly formalize the assumptions underlying Theorem 3.1 and 3.2 to clear up any ambiguity.
> - **Prior Assumption**: We assume a data-independent, cardinality-aware prior $P(\Delta) \propto \exp(-\alpha|\Delta|)$, which mathematically formalizes our preference for smaller, minimal premise sets.
> - **Posterior Definition ($Q$)**: The posterior distribution $Q$ is defined by the LLM's autoregressive generation probability. Crucially, in our practical algorithm, $Q$ is defined by conditioning this generative probability on the event that the logical chain successfully passes the solver's validation.
> - **Empirical Risk**: We assume that the logic solver is sound, meaning it drastically reduces empirical risk to near zero for validated reasoning paths.
>
> >**Q1: Why it has to be a minimal set? Can I just provide a lot of factual knowledge that contain all I need, and then make inference? Would that reduce time costs?**
>
> **Response:** Thanks for the helpful feedback. The “minimal set” requires selecting premises or knowledge that are both sufficient and necessary to achieve the logical reasoning path, while excluding all other redundant information. **Without a minimal set constraint, LLMs may generate many insufficient or unnecessary knowledge claims during completion, which does not benefit the reasoning path completion.** Instead, the excessive and redundant information makes the model prone to hallucinations and causes unnecessary token consumption. The empirical validation results are provided below.
>
> **We conducted experiments using a maximal set for each completion.** Compared to the current minimal set configuration, **the average time cost increased by 59.57%.** The table below shows **the average token count, number of API calls, and performance** for both the minimal set and the maximal set.
>
> ||Avg Token|Avg API Calls|Acc|Recall|F1|
> |-|-|-|-|-|-|
> |minimal set|3106.92|4.77|90.81|94.12|91.82|
> |maximal set|3572.94|5.31|85.28|83.73|86.15|
> ||
>
> Additionally, based on RAG and with a fixed Top-k, we **input different amounts of factual knowledge at the initial stage**, and the results are as follows:
>
> |Fact num|Acc|Recall|F1|
> |-|-|-|-|
> |0 |90.81|94.12|91.82|
> |3|86.59|85.84|87.52|
> |5|86.95|84.37|87.01|
> |10|85.24|83.37|85.86|
> |15|85.24|83.02|85.19|
> |20|85.08|83.02|85.06|
> ||
>
> Based on the above results, we found that:
> -    Compared to other settings, **the minimal set yields higher accuracy while significantly reducing both time and token cost.**
> -    In RAG methods, **increasing the amount of factual knowledge actually leads to a decrease in accuracy.**
>
> >**Q2: For more complicated logical reasoning cases, is it possible that the system iteratively updates the premises up to K_max but still fail in logic solver? How the proposed framework deals with it?**
>
> **Response:** Thanks for the suggestion. Yes, this does occur. When the iteration count of premise updates reaches K_max but the logic solver still fails, **it indicates that the sample may not be missing any premises, or that the logic solver is unable to execute the reasoning for this sample**. Therefore, in this case, we will **adopt the Chain-of-Thought (CoT) strategy, prompting the LLM to reason directly without premise supplementation**, and output the final answer.
>
> >**Limitations: The paper does not include a discussion of its limitations or the potential negative societal impacts of the proposed work.**
>
> **Response:** We recognize two limitations of this work. First, the symbolic language used is limited to first-order logic (since we used only Prover9 here). If other symbolic languages are better suited to the target real-world samples, we can substitute Prover9 with other corresponding logic solvers, such as Z3. Second, in some highly specialized or low-resource domains, general LLMs may not be effective in the fact verification. In such cases, we could consider using domain-specific LLMs as alternatives.
>
> ---
>
> Please let us know if you’d like further illustration, thank you!

---

> > ### Author Rebuttal · Reviewer_Pg5E · 2026-04-03
> >
> > Thank you for your response to my question. I will maintain my current score.

---

> > > ### Author Response · Authors · 2026-04-03
> > >
> > > We really appreciate that your concerns have been adequately addressed and we are truly grateful for your dedicated time and constructive comments!

---

### Decision · Program_Chairs · 2026-04-30

**Decision:**

Accept (regular)

**Comment:**

This paper explores logical reasoning without the "closed-world assumption" where it is assumed that all the necessary premises are explicitly provided for the reasoning system according to which a conclusion could be made. The paper considers that in reality, however, necessary premises are sometimes missing, or are considered as implicit knowledge that are not recorded explicitly, making it impossible to draw conclusions with broken logical reasoning chains. To enable the LLM-based reasoning systems to be able to conduct logical reason under such open-world setting, the paper proposes an iterative framework based on LLM-based fact verification and logic verifier, where the logic verifier identifies the logical gap, and the LLM provides additional facts/premises to make the logical reasoning complete. The paper justifies their proposed framework from empirical experiments in some simple settings.

The paper is thought provoking and novel, however with clear limitations in terms of the generality of the approach, scalability, assumed FOL parseability and comparison with RAG. The authors should consider strengthening the paper with this feedback for the next version and explicate these limitations in the main paper.